**The outflow of Asian biomass burning carbonaceous aerosol into the UTLS in spring:**

**Radiative effects seen in a global model**

Prashant Chavan[1,2], Suvarna Fadnavis[1*], Tanusri Chakroborty[1], Christopher E. Sioris[3],
Sabine Griessbach[4], Rolf Müller[5]

[1]Indian Institute of Tropical Meteorology, Center for climate change, MoES, India
[2]Savitribai Phule Pune University, Pune, India,
[3]Air Quality Research Division, Environment and Climate Change, Toronto, Canada
[4]Forschungszentrum Jülich GmbH, Jülich Supercomputing Center, Jülich, Germany,
[5]Forschungszentrum Jülich GmbH, IEK7, Jülich, Germany
Corresponding author email: suvarna@tropmet.res.in

**Abstract**

Biomass burning (BB) over Asia is a strong source of carbonaceous aerosols during spring. From ECHAM6-HAMMOZ model simulations and satellite observations, we show that there is an outflow of Asian BB carbonaceous aerosols into the Upper Troposphere and Lower Stratosphere (UTLS) (black carbon: 0.1 to 6 ng m$^{-3}$ and organic carbon: 0.2 to 10 ng m$^{-3}$) during the spring season. The model simulations show that the greatest transport of BB carbonaceous aerosols into the UTLS occurs from the Indochina and East Asia region by deep convection over the Malay peninsula and Indonesia. The increase in BB carbonaceous aerosols enhances atmospheric heating by 0.001 to 0.02 K d$^{-1}$ in the UTLS. The aerosol-induced heating and circulation changes increase the water vapour mixing ratios in the upper troposphere (by 20-80 ppmv) and in the lowermost stratosphere (by 0.02-0.3 ppmv) over the tropics. Once in the lower stratosphere, water vapour is further transported to the South Pole by the lowermost branch of the Brewer-Dobson circulation. These aerosols enhance the in-atmosphere radiative forcing (0.68±0.25 W m$^{-2}$ to 5.30±0.37 W m$^{-2}$), exacerbating atmospheric warming but produce a cooling effect on climate (TOA: -2.38±0.12 W m$^{-2}$ to -7.08±0.72 W m$^{-2}$). The model simulations also show that Asian carbonaceous aerosols are transported to the Arctic in the troposphere. The maximum enhancement in aerosol extinction is seen at 400 hPa (by 0.0093 km$^{-1}$) and associated heating rates at 300 hPa (by 0.032 K d$^{-1}$) in the Arctic.

## 1.  Introduction

There is growing concern about increasing aerosol amounts over South and East Asia, not only because of its contribution to air pollution and its harmful health effects (Chen et al., 2017; Thomas et al., 2019), but also because of its impact on the hydrological cycle (Meehl et al., 2008). Biomass burning (BB) accounts for ~60 % of the total aerosol optical depth (AOD) globally (Cheng et al., 2009; Streets et al., 2003). It is one of the major sources of large carbonaceous aerosol (Ni et al., 2019). BB is responsible for the major fraction of global mean emissions of black carbon (BC, ~59%) and organic carbon (OC, ~85 %) (Bond et al., 2013).

In Asia, China (25 %) is the largest contributor to the global BB aerosol emissions, followed by India (18 %), Indonesia (13 %), and Myanmar (8 %) (Streets et al., 2003). Among the sources, forest burning (anthropogenic and natural) contributes 45 %, burning of crop residues in the field 35 %, and burning grassland and savannah 20 % to the total BB aerosols in Asia (Streets et al., 2003). Asia emits a substantial amount of BC (~ 0.45 Tg yr$^{-1}$) and OC (~3.3 Tg yr$^{-1}$) from BB (Streets et al., 2003).  These are significant fractions of the global BB emissions of BC (~2.8–4.9 Tg yr$^{-1}$) and OC (~31–36 Tg yr$^{-1}$), respectively (Andreae, 2019). Recently, Wu et al. (2018) and Singh et al. (2020) reported ~83 % of the carbonaceous aerosol mass is emitted from open fires over South and East Asia. Within Asia, BB carbonaceous aerosol emissions from East Asia (BC: 110 Gg, OC: 730 Gg) are larger than over India (BC: 83 Gg, OC: 650 Gg) and the Indochina region (BC: 40 Gg, OC: 310 Gg) (Streets et al., 2003).

Biomass burning over Asia shows a strong seasonal cycle peaking in spring (Streets et al., 2003). Our analysis of MODIS fire counts over Asia also shows a pronounced peak in spring (Fig. 1a). The carbonaceous aerosols emitted from BB also peak in spring over Indochina, South Asia, and East Asia regions (Fig. 1b). These aerosols will affect the regional radiative forcing. The literature shows that aerosols emitted from BB in spring produce a significant negative radiative forcing at the top of the atmosphere (TOA) and at the surface, but in-atmospheric radiative forcing (TOA - surface) is positive over Asia (Wang et al., 2007; Lin et al., 2014; Singh et al., 2020).

Deep convection occurs over the Bay of Bengal, the South China Sea, and Malay Peninsula during the spring and monsoon seasons (Randel et al., 2010; Fadnavis et al., 2013; Murugavel et al., 2012) that may transport Asian boundary layer pollutants to the UTLS. Numerous airborne measurements show evidence of carbonaceous aerosol in the upper troposphere over Asia and adjoining outflow regions during spring and monsoon seasons, e.g., measurements from the Civil Aircraft for Regular Investigation of the Atmosphere Based on an Instrument Container (CARIBIC) campaign in 2004, Stratospheric and upper tropospheric processes for better climate predictions (StratoClim) in 2017, Aerosol Radiative Forcing in East Asia (A-FORCE) in 2009, and Transport and Chemical Evolution over the Pacific (TRACE-P) in 2001 (Nguyen et al., 2008; Pozzoli et al., 2008; Oshima et al., 2012; Weigel et al., 2020; Brunamonti et al., 2018; Hanumanthu et al., 2020). There may be a significant contribution from BB to the observed carbonaceous aerosols in the UTLS, since BB accounts for ~59 - 80 % of the carbonaceous aerosols globally (Bond et al., 2013) and being fine-grained, these aerosols have long atmospheric residence times. Transport of Australian wildfire smoke into the stratosphere (~35km) is seen in satellite observations (Khaykin et al., 2020). The balloon-borne, lidar, and satellite observations showed pyro-

cumulonimbus events that injected smoke from Canadian forest fires into the stratosphere
in August 2017 (Peterson et al., 2018; Hooghiem et al., 2020; Lestrelin et al., 2021). The
carbonaceous aerosols were transported to the upper troposphere and produced significant
heating locally (Fadnavis et al., 2017a). The heating of the upper troposphere induces an
amplification of the vertical motion in the troposphere (Fadnavis et al., 2017b; Hooghiem,
et al., 2020).

Numerous studies show the transport of boundary layer aerosols from Asia to the lower
stratosphere during the monsoon season (Randel et al., 2010; Fadnavis et al., 2013).
However, transport of Asian aerosol pollution into the UTLS during the spring season is not
reported hitherto when the deep convection occurs over the Malay peninsula (Chang  et al.,
2005) and Indonesia, and when biomass burning aerosol emissions show a peak (Streets et
al., 2003; Fig. 1). In this study, we address these unexplored science questions (1) transport
pathways of Asian BB aerosols to the lower stratosphere during the spring season, (2)
impacts of Asian BB carbonaceous aerosols on the lower stratosphere. For this purpose, we
employ the state-of-the-art ECHAM6-HAMMOZ chemistry-climate model. The model is
evaluated against satellite (MODIS) and ground-based remote sensing (AERONET). The
paper is organized as follows: satellite data, ground-based data and the experimental set-up
are described in section 2. Section 3 comprises a discussion on the distribution of fires and
model evaluation; results are discussed in section 4; conclusions are given in section 5.





## 2. Model simulations and satellite observations

## 2.1 Model description and experimental set-up

The fully coupled chemistry-climate model ECHAM6.3–HAM2.3 is used in this study. It comprises the general circulation model ECHAM6 coupled to the aerosol sub-module "Hamburg Aerosol Model (HAM)" (Stier et al., 2005). HAM predicts the evolution of sulfate (SU), BC, OC, particulate organic matter (POM), sea salt (SS), and mineral dust (DU) aerosols. The size distribution of the aerosol population is described by seven lognormal modes with prescribed variance in the aerosol module (Stier et al., 2005). The anthropogenic and fire emissions were obtained from the ACCMIP-II (Emissions for Atmospheric Chemistry and Climate Model Intercomparison Project) emission inventories and are interpolated for the period 2000 - 2100 by using Representative Concentration Pathway 4.5 (RCP4.5) (Lamarque et al., 2010; van Vuuren et al., 2011). The biomass burning emissions dataset represent average conditions of the decade (Tegen et al., 2019). It should be noted that inter-annual variability of biomass burning is not considered in our simulations. Injection heights of biomass burning emissions are documented by Val Martin et al. (2010). The majority (75%) of the emissions are evenly distributed within the planetary boundary layer (PBL) with 17% in the first model level above the planetary boundary layer and 8% in the second model level above the planetary boundary layer (Tegen et al., 2019). Biogenic emissions are derived from MEGAN (Guenther 1995). In the model, biogenic OC is directly inserted via emissions. Secondary organic aerosol (SOA) emissions are as described by Dentener et al. (2006).

The model simulations are performed at a T63 spectral resolution corresponding to 1.875°×1.875° horizontal resolution, while 47 hybrid σ-p levels provide the vertical

resolution from the surface up to 0.01 hPa. The model has 12 vertical levels in the UTLS

(300 to 50 hPa). The simulations have been carried out at a time step of 20 min. Atmospheric

Model Inter-comparison Project (AMIP) monthly varying sea surface temperature (SST)

and sea ice cover (SIC) were used as lower boundary conditions. We performed two sets of

emission sensitivity experiments; in one set of the simulations, the aerosol emissions from

biomass burning were kept on (referred to as BMaeroon simulations) and in another set of

the simulations, the aerosol emissions from biomass burning were kept off (referred to as

BMaerooff simulations). To assess the uncertainty caused by model imperfections, we

adopted an ensemble mean approach (with ten ensemble members) for the above two

experiments. Ten spin-up simulations were performed from 1-10 January 2012 up to 28

February 2013 to generate stabilized initial fields for the ten ensemble members. Emissions

were the same in each of the ten members during the spin-up period. In the BMaerooff

simulations (ten ensemble members each), the biomass burning aerosols were switched off

since 1 March 2013. The BMaeroon and BMaerooff simulations ended on 31 December

2013. To investigate the effects of biomass burning aerosol emissions in spring (i.e., since 1

March 2013), we analyze the difference between BMaeroon and BMaerooff simulations for

the spring season in 2013. The uncertainty estimates in simulated radiative forcing, heating

rates, and aerosol extinction coefficient are obtained from the difference between the mean

of (a) the ten-members for BMaeroon and (b) the ten-members for BMaerooff. Both sets

were generated from initial conditions with start times shifting by a day over the ten days

period of 1-10 January. The year 2013 was chosen for the analysis as this was a neutral year

without a pronounced El Niño or Indian Ocean Dipole oscillation. Such large-scale coupled

atmosphere–ocean oscillations substantially affect the transport processes to the UTLS

(Fadnavis et al., 2017a, 2019).

**2.2 MODIS fire counts and aerosol optical depth**

In order to study spatio-temporal variations in the biomass burning activity, we analysed the Terra/Aqua combined daily active fire location data (product mcd14dl) from the Moderate Resolution Imaging Spectroradiometer (MODIS) (https://firms.modaps.eosdis.nasa.gov/download/) onboard Terra and Aqua (Earth Observing System). This MODIS collection-6, Level-2 global data are processed by NASA's Land, Atmosphere Near real-time Capability for EOS (LANCE) Fire Information for Resource Management System (FIRMS), using swath products (MOD14/MYD14). The thermal anomaly / active fire represents the centre of a 1 km pixel that is flagged by the MODIS MOD14/MYD14 Fire and Thermal Anomalies algorithm as containing one or more fires within the pixel (Giglio et al., 2003). The fire detection algorithm uses the strong mid-infrared (IR) emissions from the fires (Matson and Dozier 1981) and is based on the brightness temperatures derived from MODIS at the 4 and 11-μm channels. The retrieval algorithm classifies fire pixels in three categories: low confidence (0 – 30 %), nominal confidence (30 – 80 %), and high confidence (>80 %). This confidence limit allows the rejection of false fires (Giglio, 2015). Here, data with high or nominal confidence (≥70 %) are used.

For information on aerosol, we used monthly mean data from MODIS Terra (MOD08 M3 V6.1) at 1°×1° horizontal resolution to study AOD variability over the Asian region during spring 2013. MODIS Terra measures radiance emanating from the surface and the atmosphere and provides images in 36 spectral bands between 0.415 and 14.235 μm, with a spatial resolution varying from 250 m to 1 km (Mhawish et al., 2019). Terra MODIS

MOD08_M3 (V6.1) aerosol products (i.e., AOD) are retrieved using the Deep Blue (DB)
algorithm. The algorithm calculates the column aerosol loading at 0.55 μm over land and
ocean.

**2.3 Multi-Angle Imaging Spectroradiometer (MISR), Aerosol Robotic NETwork**
**(AERONET) and Optical Spectrograph and InfraRed Imaging System (OSIRIS)**
**observations**

The AOD retrievals from the Multi-Angle Imaging Spectroradiometer (MISR) at 550
nm wavelength and the Aerosol Robotic NETwork (AERONET) sunphotometer during
spring 2013 are also used for comparison with the model simulations. Details of MISR are
available at https://misr.jpl.nasa.gov/getData/accessData/ and AERONET at
https://aeronet.gsfc.nasa.gov/. AERONET AOD observations are obtained at different
stations in the Indochina region (Myanmar: 16.86°N - 96.15°E, Vientiane: 17.99°N-
102.57°E, Siplakorn University: 13.81°N-100.04°E, Ubon-Ratchathani: 15.24°N -
104.87°E), South Asia (Gandhi college: 25.81°N - 85.12°E, Lumbini: 27.49°N-83.28°E,
Kathmandu Bode: 27.68°N -85.39°E, Dhaka University: 23.72°N - 90.39°E), East Asia
(Nghia-Do: 21.04°N - 105.80°E, Hong Kong Polytechnic University: 22.30°N - 114.18°E

196    ).


We compared simulated aerosol extinction coefficient vertical profiles with observations from
Optical Spectrograph and InfraRed Imaging System (OSIRIS) on-board the Odin satellite
(Bourassa et al., 2012). We used version 7.0 vertical profiles of aerosol extinction at 750
nm for March-May 2013 (https://research-groups.usask.ca/osiris/data-
products.php#Download). The limb scatter measurements from OSIRIS show good

agreement with Stratospheric Aerosol and Gas Experiment (SAGE) II and Scanning

Imaging Absorption spectrometer for Atmospheric Chartography (Rieger et al., 2018). To

understand convective activity in spring 2013, we also analyzed Outgoing Longwave

Radiation (OLR) data for March - May 2013 from the National Center for Environmental

Prediction                                                  (NCEP)                                                  re-analysis-2

(https://psl.noaa.gov/data/gridded/data.ncep.reanalysis2.pressure.html).

**3. Distribution of fires and model evaluation**

**3.1 Seasonal distribution of fires over Asia**

In this section, we discuss the seasonal variability of fire activity in Asia. The fire counts

peak over Asia (10°S - 50°N, 60°E - 130°E) in the spring season.  Figure 1a-b shows that

fires are clustered over three sub-regions (1) Indochina region (91°E - 107°E, 10°N - 27°N)

(numbers of fire counts: 80694), (2) East Asia (108°E - 123°E, 22°N - 32°N), (numbers of

fire counts: 4770), (3) South Asia (65°E - 90°E, 8°N - 32°N) (numbers of fire counts: 14223)

(Fig. 1b). Fire counts over the three sub-regions peak in spring although the month varies,

e.g., fire counts over East Asia show a peak in March, Indochina region in March-April, and

South Asia in May (Fig. 1a). The fire counts over South Asia show a secondary peak in

October. In agreement with our results, Bhardwaj et al. (2016) also reported high fire activity

in spring and the lowest fire activity during the monsoon (June–September) in the 2003-

2013 time frame. Streets et al. (2003) reported that higher fire counts during the spring

season over South Asia and East Asia are attributed to enhanced crop burning activity. Over

the Indochina region, high fire counts are associated with forest fires along with crop

burning. Intense biomass burning activity over Asia during the spring season is also reported

by Zhang et al. (2020). Hence, we provide further analysis in spring.

## 3.2. Model evaluation

We compare simulated AOD (averaged for spring from BMaeroon simulations) with MODIS, MISR, and AERONET. Figure 2 (a-c) shows large AOD over the regions: Indochina (MODIS: ~0.4 to 0.8, MISR: 0.27 to 0.6; model: 0.27 to 0.5), East Asia (MODIS: 0.5 to 1.3, MISR: 0.27 to 1, model: 0.5 to 1.4), and the Indo-Gangetic plain in south Asia (23°N -30°N, 75°E - 85°E) (MODIS: 0.24 to 0.8, MISR: 0.24 to 0.5, model: 0.3 to 0.6). The MISR AOD is comparatively less than MODIS AOD over all three study regions (Fig. 2a-b). There are differences in the spatial distribution of AOD among MODIS, MISR and the model. Over East Asia, the model overestimates AOD relative to MISR (by 0.24) and MODIS (by 0.1). Over Indochina, the model shows an underestimation compared to MISR (by 0.1) and MODIS (by 0.2). The simulated AOD is over-estimated over the Indo-Gangetic plain in comparison with MISR (by 0.08) and underestimated compared to MODIS (0.2). The simulated AOD is underestimated south of 13°N compared to MISR and MODIS (MODIS: 0.4 to 0.7, MISR: 0.4 to 0.6, model: 0.21 to 0.3) and overestimated over central India (lat: 20° - 28°N lon: 75°E - 88°E) compared to MODIS and MISR (MODIS: 0.16 to 0.4, MISR: 0.21 to 0.3, model: 0.3 to 0.5). These issues may be due to a higher amount of dust emission in the model over West Asia that is transported to India. In the past, a number of papers reported that transport of dust occurs from west Asia to the Indo-Gangetic plain and the Tibetan Plateau region during spring (Lau and Kim 2006; Fadnavis et al., 2017b, Fadnavis et al., 2021a). Simulated AOD is also overestimated over the Tibetan Plateau and East Asian region (MODIS: 0.21 to 1.0, MISR: 0.16 to 0.6, model: 0.27 to 1.2). The distribution of dust AOD also shows high amounts over these regions (See Fig. S1). This indicates that higher amounts of dust over the Tibetan Plateau and the East Asia region cause overestimation of AOD there. Tegen et al. (2019) also reported that in ECHAM6–

HAMMOZ simulations the AOD is overestimated over East Asia in comparison with MISR.
The model simulations underestimate the AOD over the Himalayas in comparison with
MODIS (MODIS: 0.24 to 0.3, MISR: 0.1 to 0.21, model: 0.1 to 0.3). It should be noted that
dust emission/parameterization is the same in both BMaeroon and BMaerooff simulations.

Further, we compare simulated AOD with ground-based measurements at ten AERONET
stations during spring 2013 (Figure 2d). Model results were sampled at each station at the
same time. Comparison with AERONET observations also shows that the model
underestimates AOD over all the stations. The simulated AOD (0.54) shows the highest
underestimation at Nghia Do (21.04°N - 105.80°E) in East Asia and the lowest
underestimation at Gandhi college (25.81°N - 85.12°E) in the Indo-Gangetic plain, where
the simulated 550 nm AOD is 0.57.

The differences in the magnitude of AOD between model, satellite remote sensing (MISR,
MODIS), and ground-based AERONET observations may be caused by various factors;
e.g., satellite remote sensing of AOD exhibits biases over certain surface types. The
differences between MISR and MODIS may be due to differences in their calibration,
algorithm assumptions, or the aerosol models in the lookup tables used in the retrieval
algorithms (Addou et al., 2005; Choi et al., 2019). There are uncertainties in the model
emission inventories (Fadnavis et al., 2013, 2017, 2019).

The vertical distribution of simulated aerosol extinction coefficient profiles (BMaeroon)
averaged over the BB burning region (10ºN - 30ºN) are compared with OSIRIS observations
in spring 2013 (Fig. 2e-f). Our model could simulate vertical variations similar to those
observed by OSIRIS. A plume rising from 90ºE - 120ºE extends to 16 km is also evident in
the OSIRIS data although the model underestimates the aerosol extinction coefficient by
0.0002 - 0.0003 $km^{-1}$. The sign of the difference is consistent with the slightly shorter
wavelength of the OSIRIS extinction measurements. This underestimation may also be due
to uncertainties in the model due to emission inventory and transport processes in the model.
It should be noted that there may be biases in OSIRIS measurements due to assumptions
made on the aerosol size distribution and chemical composition (Bourassa et al., 2012).

**4. Results**
**4.1 Impact of biomass burning on Aerosol Optical Depth (AOD)**

Figure 3 (a) shows the distribution of anomalies in simulated AOD (BMaeroon-

BMaerooff). It shows enhanced AOD anomalies over the Indo-Gangetic plain (~0.22 to 0.8),
the Tibetan Plateau and the north eastern parts of East Asia (~0.3 to 1.2). The distribution of
anomalies in dust AOD shows high amounts over these regions. It indicates that dust
enhancement over the Indo-Gangetic plain (~0.22 to 0.8), the Tibetan Plateau and the
northeastern parts of East Asia (0.8 to 1) (Fig. 3b) causes enhancement in AOD there. The
simulated dust anomalies and circulation patterns also show transport of enhanced dust from
West Asia to North India and the Indo-Gangetic plain region in the lower troposphere (Fig.
3b and Fig. S2a). Dust is also transported from Tibetan Plateau-East Asia region to North
India in the mid/upper troposphere (Fig. S2b). The enhanced dust transport from west Asia
and Tibetan Plateau-East Asia region to South Asia is induced by atmospheric heating
generated by biomass burning carbonaceous aerosols (discussed in section 4.4). This
atmospheric heating leads to enhanced dust emission over the respective desert regions.
Dust being absorptive in nature contributes to a further increase of the atmospheric heating.
The heating led to a formation of a low pressure zone over East India in the lower
troposphere (900 hPa) (Fig. 3b) and the Bay of Bengal and Myanmar in the mid-troposphere
(500 hPa) (Fig. S2b and Fig. 7b). These circulation changes further enhanced the dust
transport from west Asia and the Tibetan Plateau-East Asia region to South Asia.

Figure 3c shows the spatial distribution of the AOD for carbonaceous aerosols (BC+OC). The
changes in concentration of total column carbonaceous aerosols are shown in Fig. S3a.
Figures 3c and S3a show increases in aerosols over Indochina (AOD: +0.04-0.07,
concentration: +40 - 80 %), Indo-Gangetic plain (AOD: +0.014-0.03, concentration: +10-
50 %) and East Asia (AOD: +0.018-0.04, concentration: +20 - 60 %). It is evident that
anomalies of carbonaceous aerosols AOD over the Indo-Gangetic plain and East Asia are
comparatively lower than over the Indochina region. In agreement with our results, Wang et
al. (2015) also reported an abundant mixture of BC and OC particles due to BB over the
Indochina region in spring 2014. Our model simulations show that the contribution of BB-
emitted OC to AOD (Indochina 16 to 35 %; East Asia: 4 to 12 %; South Asia: 0.8 to 4 %) is
higher than that of BB-emitted BC (Indochina: 1.8 to 6 %; East Asia: 0.8 to 1.4 %; South
Asia: 0.2 to 0.8 %) (Fig. S3b-c). Figure 3c also shows high amounts of carbonaceous
aerosols over the western Pacific, which may be due to transport from the Indochina region
by westerly winds (discussed later in subsection 4.3).

**322  4.2. Impact of BB carbonaceous aerosol on radiative forcing**


The carbonaceous aerosols emitted from biomass burning may significantly change

radiative forcing by absorption and attenuation of solar and terrestrial radiation (Schill et
al., 2020). The anomalies (averaged for spring) in net radiative forcing show negative
radiative forcing at the surface and top of the atmosphere (TOA) over South Asia (surface:
-5.08±0.44 W m$^{-2}$; TOA: -4.39±0.26 W m$^{-2}$), Indochina region (surface:-7.68±0.45 W m$^{-2}$;
TOA: -2.38±0.12 W m$^{-2}$) and East Asia (surface:-10.81±0.63 W m$^{-2}$; TOA: -7.08±0.74 W
m$^{-2}$) (Fig. 4). The estimates of in-atmosphere radiative forcing show positive anomalies over
south Asia (0.68±0.25 W m$^{-2}$), Indochina region (5.30±0.37 W m$^{-2}$), and East Asia
(3.73±0.20 W m$^{-2}$), indicating an atmospheric warming. In agreement with our study, a
number of studies showed a negative radiative impact at the TOA and surface, but positive
in-atmosphere radiative forcing due to BC and OC aerosols over the Indochina region. For
example, Lin et al. (2014) reported a radiative forcing of -4.74 W m$^{-2}$ at the TOA, -26.85 W
m$^{-2}$ at the surface, thus +22.11 W m$^{-2}$ in-atmosphere. Wang et al. (2007) estimated a radiative
forcing of -1.4 to -1.9 W m$^{-2}$ at the TOA and -4.5 to -6 W m$^{-2}$ at the surface, yielding +2.6
W m$^{-2}$ in-atmosphere during March 2001. Singh et al. (2020) also reported a radiative
forcing at the TOA of -1.91 W m$^{-2}$ and -42.76 W m$^{-2}$ at the surface and 40.85 W m$^{-2}$ in-
atmosphere over Myanmar.

**4.3. Transport of biomass burning aerosol into the upper troposphere and lower**
**stratosphere**

The stepwise evolution of the Asian summer monsoon begins in spring and contributes a
significant amount of rainfall to the total annual precipitation over China (25 – 40 %) and
over South Asia (~11 - 20 %) due to deep convection over the Bay of Bengal, Tibetan Plateau
and South China Sea (Guhathakurta and Rajeevan, 2008; Li et al., 2016).  The distribution
of outgoing long-wave radiation (OLR) from NCEP reanalysis data during the spring season
confirms that deep convection occurs over the maritime continent that extends to the South
China Sea, Bay of Bengal, Malay Peninsula and Indonesia (Fig. 5a). Our model simulation
shows a distribution of OLR similar to the observations, although OLR is overestimated in

the model (Fig. 5b). Figure 5(c)-(d) shows the combined distribution of Cloud Droplet Number Concentration (CDNC), Ice Crystal Number Concentration (ICNC), and vectors of the resolved circulation, which exhibit a strong upwelling in equatorial Asia (10°N - 20°N, 85°E - 140°E, Fig. 5c-d). This upwelling associated with deep convection may transport pollutants from the boundary layer into the UTLS.

We analyzed the vertical distribution of simulated anomalies (BMaeroon - BMaerooff) of BB carbonaceous aerosols obtained over the high fire emission regions, i.e., Indochina, South Asia, and East Asia in spring 2013 (Fig. 1b). The simulated distribution of BC aerosols (Fig. 6 a-b) and OC aerosols (Fig. 6c-d) over the Indochina region indicates an aerosol plume extending to the lowermost stratosphere. The ascent resolved in the wind vectors together with the distribution of cloud droplets and cloud ice indicate that the transport of these aerosols from the surface to the lowermost stratosphere occurs due to deep convection over the Malay peninsula and Indonesia (Fig. 5a-b). There is an enhancement of BC aerosol concentration by $0.1 - 2$ ng m$^{-3}$ (Fig. 6 a-b) and for OC by $0.2 - 5$ ng m$^{-3}$ (Fig. 6 c-d) in the UTLS (300 - 90 hPa) over the Indochina region.

In the troposphere, biomass-burning carbonaceous aerosols are transported to the Arctic (Fig. 6a and Fig. 6c). Some previous studies also show aerosol transport from South Asia and East Asia to the Arctic (Shindell et al., 2008; Fisher et al., 2011). The carbonaceous aerosols are also transported towards the Western Pacific (Fig. 6 b-d and 6 f-h). In the Pacific (140°E - 170°W), these aerosols are lifted to the UTLS. Transport of the aerosols from the Indochina region to the Western Pacific has also been reported in the past (Dong and Fu, 2015).

Further, we show the distribution of BB carbonaceous aerosol over East Asia in Figure 6 e-h.
It shows that the plume of BC and OC aerosol crosses the tropopause (BC: 0.2–6 ng m$^{-3}$ and
OC: 0.2 to 10 ng m$^{-3}$). Figures 6e and 6g also show that the aerosol plume from the equatorial
region is lifted to the UTLS associated with the Indonesian region (130ºE - 170ºE).  Similar
to the Indochina region, BC and OC aerosols also show poleward transport to the Arctic and
horizontal transport towards the Western Pacific (Figures 6f and 6h). These aerosols are
vertically transported in the western Pacific region (130ºE - 170ºE). The distribution of
anomalies of BC and OC near the tropopause (at 100 hPa) shows outflow of Asian
carbonaceous aerosols in the UTLS over equatorial Asia and Western Pacific (5°S-20°N,
70°E - 180°E) (Fig. S4).

BB in South Asia occurs in central India (70°E - 90°E, 8°N -24°N) in spring (Fig. 1 b
and Singh et al., 2017). BC and OC emissions over South Asia during the spring season are
reported in many studies (Talukdar et al., 2015; Guha et al., 2015). The vertical distribution
of anomalies of BC and OC over south Asia shows that positive anomalies of BC and OC
aerosols extend from the surface to the upper troposphere (300 hPa) (Fig. S5). CALIPSO
derived aerosol profiles in spring 2013 also show plumes reaching up to approximately 7
km (400 hPa) (Singh et al., 2020). Unlike the Indochina region, BB carbonaceous aerosols
over Indo-Gangetic plain do not reach the lowermost stratosphere during the spring season.
Hence, hereafter we focus our discussion on the transport of BB carbonaceous aerosols and
their impacts on the UTLS for Indochina and East Asia.

Further, we analyze the aerosol enhancement over the Arctic (65ºN - 85ºN) due to the
transport of Asian biomass burning BC and OC aerosols. The vertical distribution of
anomalies of aerosol extinction shows an enhancement of 0-0.0093 km$^{-1}$ in the Arctic (1000
-100hPa) with a peak at 400 hPa (Fig. 7a). Shindell et al. (2008) also showed seasonally
varying transport of South Asian aerosols to the Arctic that maximizes in the spring season.

**4.4 Impact of BB carbonaceous aerosol on heating rates**

Carbonaceous aerosols in the atmosphere produce significant heating leading to
atmospheric warming (Fadnavis et al., 2017b). We obtained anomalies in heating rates
(shortwave + longwave) due to carbonaceous aerosols (BMaeroon - BMaerooff). Figure 7b
shows the spatial distribution of anomalies in tropospheric heating rates (averaged from
surface to tropopause). It shows that carbonaceous aerosols have induced significant
tropospheric heating over the location of dense fires; Indo-China/East Asia (0.02 to 0.12 K
$d^{-1}$). Significant heating is seen namely over the Mongolian desert (0.08 - 0.12 K $d^{-1}$). The
desert region of west Asia (Pakistan, Afghanistan, Turkistan, Kazakhstan) also shows slight
heating (0.02 - 0.04 K $d^{-1}$). The heating over the desert regions is associated with enhanced
emission of dust, a positive feedback to atmospheric heating induced by the carbonaceous
aerosols (section 4.1). Heating is higher over the Mongolian desert than over west Asia due
to the vicinity of Mongolia to the location of dense fires.

Further, we show the vertical distribution of heating rates over the Indochina region and East
Asia in Figures 8a-d. It shows that enhanced BB carbonaceous aerosols have induced
enhanced heating of the atmospheric column along the pathway through which they are
transported (Fig. 6a-h). The carbonaceous aerosol emissions over the Indochina region and
East Asia produced anomalous heating of ~0.1 to 0.04 K $d^{-1}$ in the lower troposphere (1000
hPa to 400 hPa) and ~0.008 to 0.001 K $d^{-1}$ near the tropopause (200 hPa to 80 hPa).  Figure
6 a, c, e, g shows that descending winds transport BC and OC aerosols from above the

tropopause downward and southward to 20°S. The positive anomalies in heating rates of ~0.001 to 0.004 K d$^{-1}$ in the upper troposphere at ~200 hPa near 20°S may be due to heating by these aerosols. There may be dynamic changes in response to BB carbonaceous aerosol emission. The transported Asian carbonaceous aerosols and associated dynamical changes in the Arctic enhanced heating rates by 0 - 0.032 K d$^{-1}$ between 1000 - 100 hPa (Fig. 7a). Also, transport of carbonaceous aerosol to the western Pacific (Fig. 6 b, d, f, h) by the westerly winds has increased heating by 0.008 to 0.04 K d$^{-1}$ and peaks at 250 hPa (0.04 K d$^{-1}$) over the Central Pacific (170°W - 110°W).

Figure 8 (a-d) shows positive anomalies in heating rates at the tropopause. Heating in the upper troposphere enhances the vertical motion that may enhance the transport into the lower stratosphere (Gettelman et al., 2004). Carbonaceous aerosols cross the tropopause (0.1 to 5 ng m$^{-3}$) and enter the lowermost stratosphere (18°N - 24°N) (Figs. 6 a-h). The cross tropopause transport is reinforced by enhanced vertical motion (Fig. S6a-b) produced by the heating generated by the carbonaceous aerosols.

**4.5 Impact of BB carbonaceous aerosol on water vapor**

The heating produced by the biomass burning carbonaceous aerosols may affect the distribution of water vapor in the troposphere and stratosphere. Figures 9a-b show anomalies in water vapor (BMaeroon - BMaerooff) over Indochina and East Asia. An interesting feature seen in Fig. 9a-b is the enhanced transport of water vapor (an anomaly of 0.02 - 0.5 ppmv) to the South Pole through the lower stratosphere from Indochina (91°E - 107°E, 10°N - 27°N) and East Asia (108°E - 123°E, 20°N - 35°N). The tropospheric heating might have caused elevated water vapor injection into the lower-stratosphere. The water vapour in the

lower stratosphere is further transported to the South Pole by the lower branch of the
Brewer-Dobson circulation. The water vapour reaches the Antarctic within a month
indicating fast transport.

The model simulations show noticeable enhancement of water vapor (0.4 to 1.6 ppmv) in
the northern tropics near the tropopause (150 hPa) and by 0.2 - 0.7 ppmv in the Arctic lower
stratosphere (150 hPa) (Fig. 9c). In the tropical lower stratosphere, it is increased by 0.02 -
0.3 ppmv (Fig. 9d). Water vapor, being a greenhouse gas, amplifies global warming leading
to positive feedback (e.g., Riese et al., 2012; Sherwood et al., 2018, Fadnavis et al., 2021b).
The strong negative anomalies of OLR (Fig. S6c) induced by carbonaceous aerosols also
indicate the positive feedback. Fadnavis et al. (2013) also reported an increase in water
vapor in the UTLS in response to the enhancement of aerosols. Stratospheric water vapor
plays a significant role in climate change (e.g., Oman et al., 2008; Wang et al., 2020; Xie et
al., 2020).

**5. Conclusions**

A ten-member ensemble of ECHAM6.3–HAM2.3 simulations for the spring season 2013, a
neutral year, is analyzed to study the transport of carbonaceous aerosol injected by Asian
biomass burning into the UTLS and its associated impacts on radiative forcing, heating
rates, and water vapor. To validate the model simulations, we compare simulations with
observations from (1) MODIS, (2) MISR, (3) AERONET, (4) OSIRIS during spring 2013.
The observational analysis shows reasonable agreement with the model simulations.

The BB emission increases the aerosol burden (AOD) over the Indochina region by 0.14 to
0.22 (carbonaceous aerosol concentration increase of +40-80 %), India by 0.22 to 0.38
(concentration of carbonaceous aerosol: +10-50 %), and East Asia by 0.18 to 0.26
(concentration of carbonaceous aerosol: +20-60 %).Our analysis shows that deep
convection, which occurs over the Malay peninsula and Indonesia, transports carbonaceous
aerosols  from the boundary layer of the Indochina and East Asia region into the lowermost
stratosphere (BC: 0.1 to 6 ng m$^{-3}$ for BC, OC: 0.2 to 10 ng m$^{-3}$). In the UTLS, outflow occurs
over equatorial Asia and the Western Pacific (10$^{o}$S - 20$^{o}$N, 70$^{o}$E - 180$^{o}$E). Carbonaceous
aerosols originating from Asian biomass burning are also transported to the Arctic. The
maximum enhancement in aerosol extinction (by 0.0093 km$^{-1}$) is seen at 400 hPa over the
Arctic.

The enhanced carbonaceous BC and OC aerosol emitted from BB produces a negative net
radiative forcing at the surface (India: -5.08±0.44 W m$^{-2}$, Indochina: -7.68±0.45 W m$^{-2}$, and
East Asia: -10.81±0.63 W m$^{-2}$), at the TOA (India: -4.39±0.26 W m$^{-2}$ , Indochina: -2.38±0.12
W m$^{-2}$, and East Asia: -7.08±0.74 W m$^{-2}$) and positive net radiative forcing in the atmosphere
(India: 0.68±0.25 W m$^{-2}$, Indochina: 5.30±0.37 W m$^{-2}$, and East Asia: 3.73±0.20 W m$^{-2}$)
indicating atmospheric warming, but a cooling of the climate at the surface.

The changes in BB carbonaceous aerosol induce a warming in the troposphere (0.008 – 0.1
K d$^{-1}$) and in the UTLS (~0.001 to 0.008 K d$^{-1}$) over Asia. The aerosols transported to the
Arctic enhance heating by 0 - 0.032 K d$^{-1}$, peaking at 300 hPa. The outflow of the aerosols
in the UTLS over the western Pacific by the westerly winds has increased heating by 0.008
to 0.04 K d$^{-1}$. The atmospheric heating induced by Asian BB carbonaceous aerosols led to
the transport of water vapor into the lower stratosphere (0.02 - 0.3 ppmv) over the tropics.
In the lower stratosphere, water vapour is transported to the South Pole by the lower branch
of the Brewer-Dobson circulation. Water vapor, being a greenhouse gas, amplifies
atmospheric heating, leading to positive feedback (e.g., Riese et al., 2012; Sherwood et al.,
2018). Our model simulations also show a positive feedback of dust aerosol on atmospheric
heating induced by the enhancement of carbonaceous aerosols.

Furthermore, our analysis shows that Asian biomass burning carbonaceous aerosols lead to
moistening of the troposphere in the northern hemisphere and lowermost stratosphere in the
northern tropics and southern hemisphere. An increase in stratospheric water vapour is
important as it has an impact on stratospheric temperatures and thus indirectly on
stratospheric dynamics (Maycock et al., 2013). The moistening of the stratosphere produces
a positive feedback on the climate (Banerjee et al., 2019; Dessler et al., 2013).

*Acknowledgments*: The authors thank the staff of the High Power Computing Centre (HPC)
in IITM, Pune, India, for providing computer resources and the team members of MODIS,
MISR, and AERONET for providing data. Authors are thankful to two anonymous
reviewers for their valuable suggestions.

**Data availability:** The MODIS fire count data were downloaded from
https://firms.modaps.eosdis.nasa.gov/download.  The AOD data from MODIS Terra can be
downloaded                                               from
https://ladsweb.modaps.eosdis.nasa.gov/archive/allData/61/MODATML2/
The AOD data from MISR were obtained from
https://misr.jpl.nasa.gov/getData/accessData/. The AERONET data were obtained from
https://aeronet.gsfc.nasa.gov/. Data of NCEP reanalysis-2 outgoing longwave radiation
(OLR)                        were                   obtained                   from
https://psl.noaa.gov/data/gridded/data.ncep.reanalysis2.pressure.html. The OSIRIS aerosol
extinction coefficient can be downloaded from https://research-groups.usask.ca/osiris/data-
products.php#Download
**Author contributions**: S. F. initiated the idea. P. C. and T. C. performed model analysis. R
M., S.G and C.E.S. contributed analysis and study design. C. E. S.and S.G. analyzed OSIRIS
data. All authors contributed to the writing and discussions of the manuscript.
**Competing Interests**: The authors declare no competing interests.

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

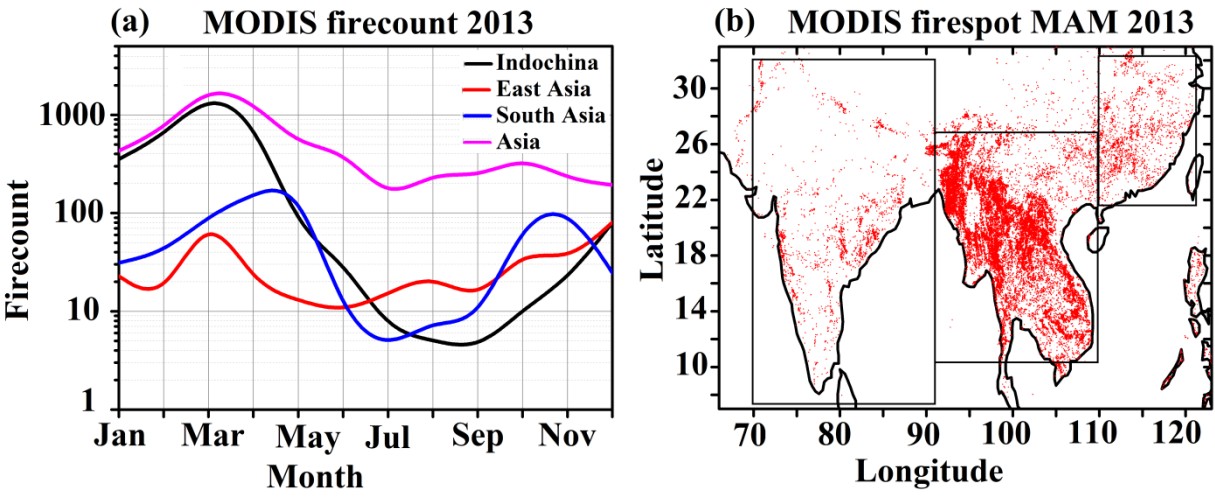


**Figure 1:** (a) Monthly mean distribution of MODIS fire counts averaged over Indochina

(91ºE - 107ºE, 10ºN - 27ºN), East Asia (108ºE - 123ºE, 22ºN - 32ºN), South Asia (70ºE -

90ºE, $8^0$N - $32^0$N) and Asia (60ºE - 130ºE, 10ºS - 50ºN) (b) Spatial distribution of fire spots

over South Asia, Indochina and East Asia averaged for spring 2013. Boxes in Figure (b)

indicate the boundaries of South Asia, Indochina, and East Asia.

816

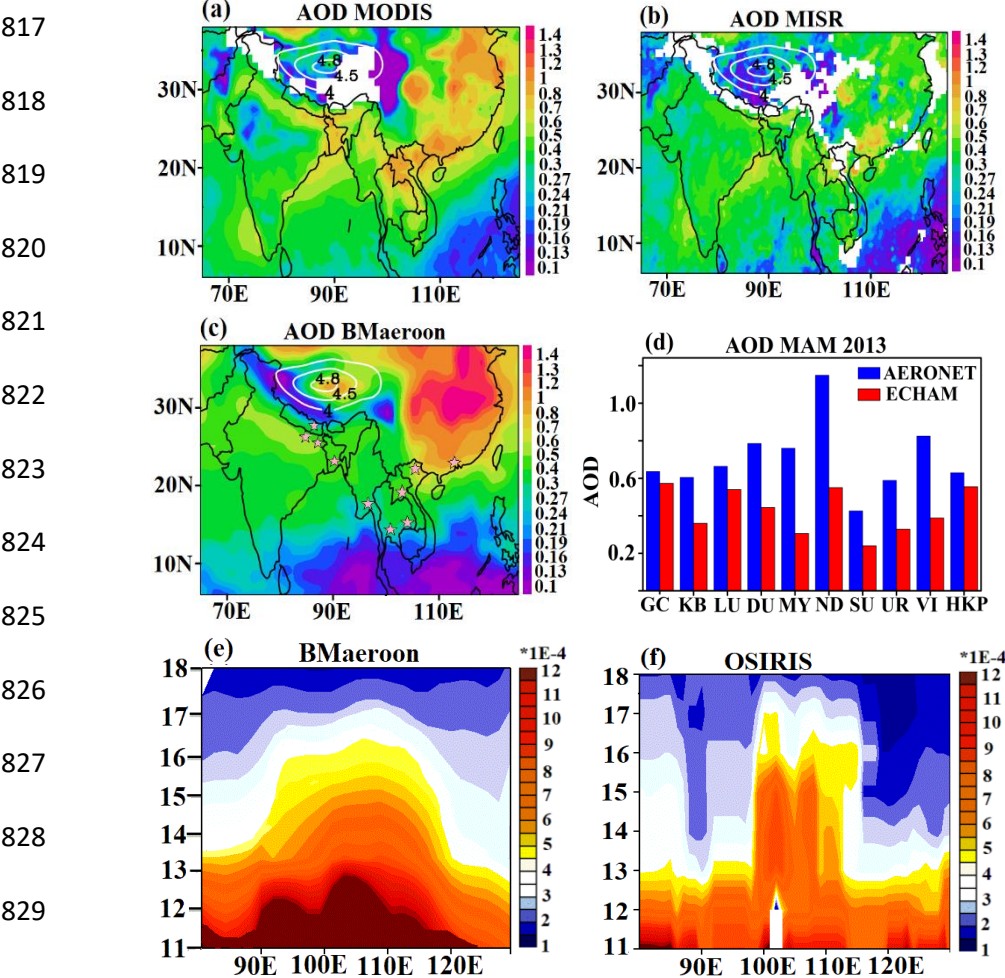

**Figure 2**: (a) Aerosol optical depth (AOD) averaged for spring 2013 from MODIS, (b) same as (a) but from MISR, (c) same as (a) but from ECHAM6 - HAMMOZ BMaeroon simulation. White contours in Fig (a)-(c) indicate the orography (km) of the Tibetan Plateau, (d) Comparison of simulated AOD (from BMaeroon) averaged for spring 2013 with AERONET observations at Gandhi college (GC; 25.81°N - 85.12°E ), Kathmandu Bode (BD; 27.68°N - 85.39°E), Lumbini (LU; 27.49°N - 83.28°E), Dhaka University (DU; 23.72°N - 90.39°E), Myanmar (MY; 16.86°N-96.15°E), Nghia Do (ND; 21.04°N - 105.80°E), Silpakorn University (SU; 13.81°N - 100.04°E), Ubon Ratchathani (UR; 15.24°N - 104.87°E), Vientiane (VI; 17.99°N - 102.57°E), Hong Kong Poly (HKP; 22.30°N – 114.18°E). (e) Simulated (BMaeroon) aerosol extinction coefficient (865 nm) (km[-1]), averaged for 12°N -30°N and spring 2013 (f) same as (e) but from OSIRIS measurements (750 nm).


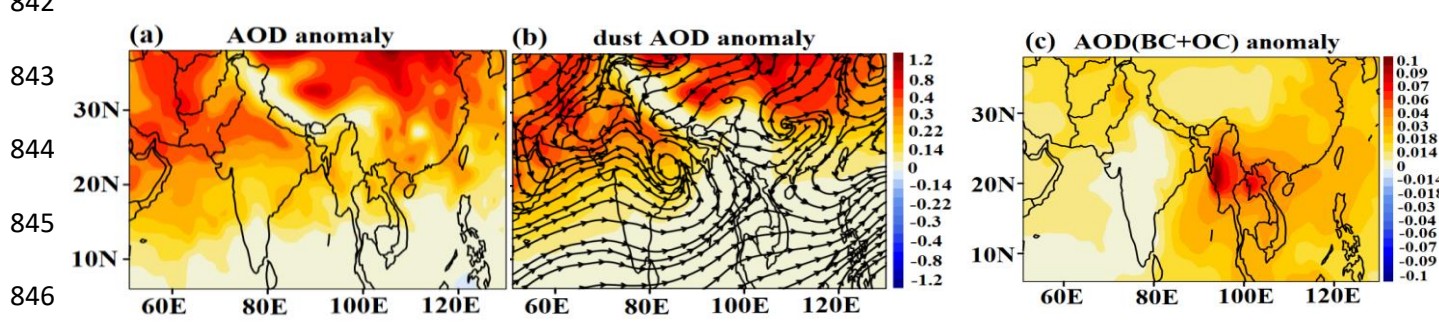






**Figure 3:** Distribution of ECHAM6-HAMMOZ simulated anomalies of (BMaeroon - BMaerooff) (a) AOD, (b) dust AOD, (c) BC-AOD and OC-AOD, together, averaged for spring 2013. Streamlines in Figure 3(b) indicate wind anomalies at 900 hPa (BMaeroon-BMaerooff).





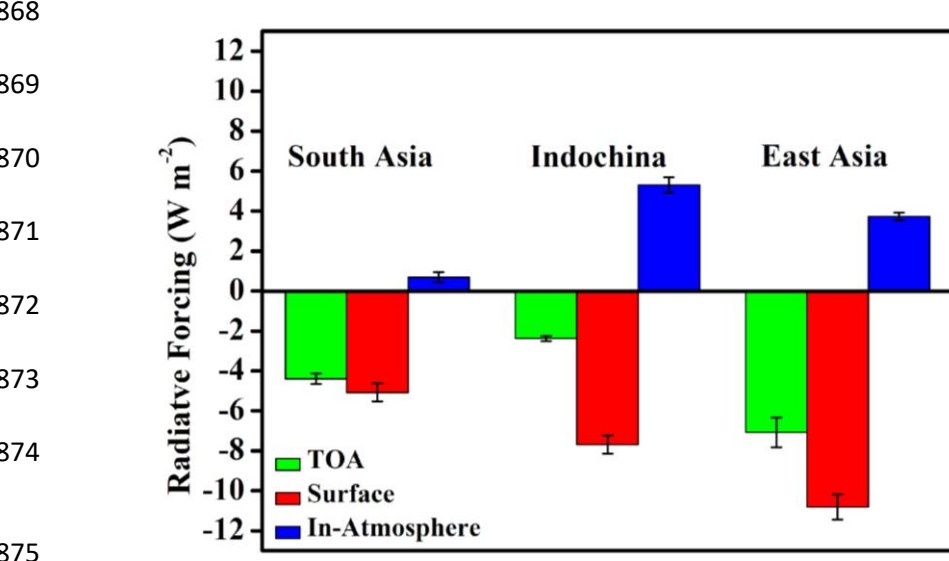

876

**Figure 4**: Anomalies of radiative forcing (W m$^{-2}$) from ECHAM6-HAMMOZ simulations

(BMaeroon - BMaerooff) at the TOA, surface, and in-atmosphere (TOA - Surface) averaged

for spring 2013 and over South Asia, Indochina, and East Asia.

880

881

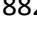

**Figure 5**: (a) Distribution of Outgoing Longwave Radiation (OLR) (W m$^{-2}$) from NCEP reanalysis-2 data averaged for spring 2013, (b) same as (a) but from the ECHAM6-HAMMOZ simulations (BMaeroon). Vertical distribution of cloud droplet number concentration (CDNC) and ice crystal number concentration (ICNC) (1 mg$^{-1}$) averaged for spring 2013 from ECHAM6-HAMMOZ simulations (BMaeroon) (c) latitude-pressure section (average for 85$^0$E-140$^0$E) and (d) longitude-pressure section (average for 10$^o$N - 20$^0$N). Vectors of the circulation (BMaeroon) are shown in Figure 5(c-d) with the vertical velocity field scaled by 300.

903

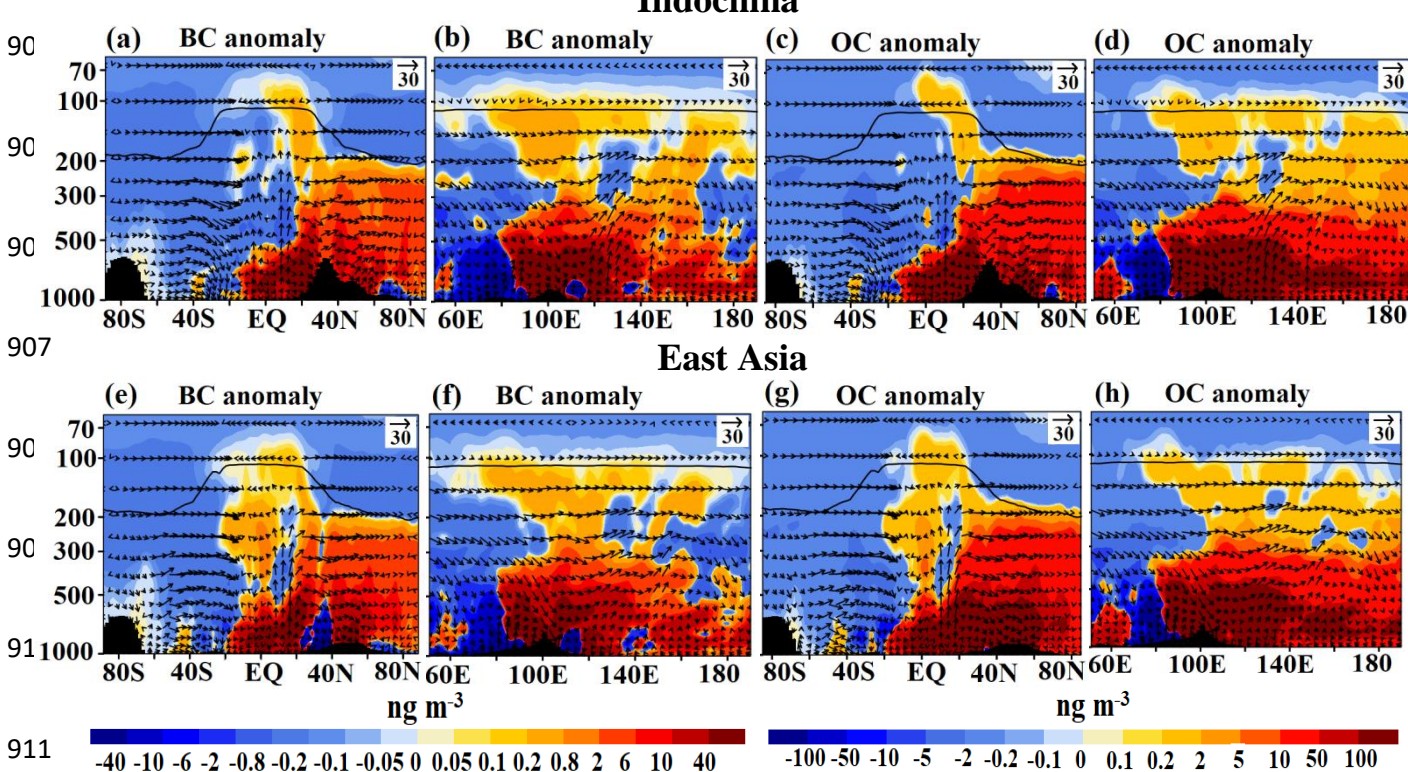

**Figure 6**: Vertical cross-section of anomalies of BC (ng m$^{-3}$) (BMaeroon – Bmaerooff) averaged for spring 2013 (a) latitude-pressure section (averaged for 91ºE-107ºE), (b) longitude-pressure section (averaged for 18ºN-24ºN). (c-d) is the same as (a-b) but for OC. (e) same as (a) but averaged over 108ºE-123ºE, (f) same as (b) but averaged for 18ºN-24ºN. (g-h) same as in (e-f) but for OC. The arrows in Figure 6(a-h) indicate winds in m s$^{-1}$ with the vertical velocity field scaled by 300. The black vertical bars show the topography and the black line indicates the tropopause.








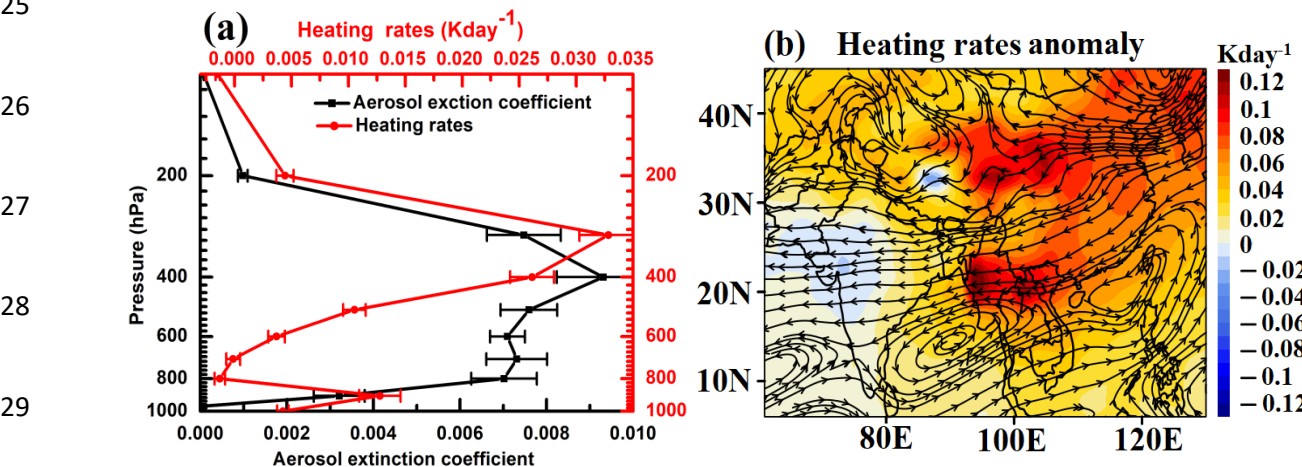

**Figure 7**: (a) Vertical profile of anomalies of extinction (km[-1]) and heating rate (K d[-1]) over

the Arctic region (65ºN-85ºN) from the ECHAM6-HAMMOZ simulations (BMaeroon -

BMaerooff). The horizontal lines indicate the standard deviation within the 10 members of

the different initial conditions, (b) spatial distribution of anomalies of heating rates (K d[-1])

(short and long wave together) averaged from the surface to the tropopause. Streamlines in

Figure 7(b) indicate wind anomalies at 500 hPa (BMaeroon- BMaerooff).









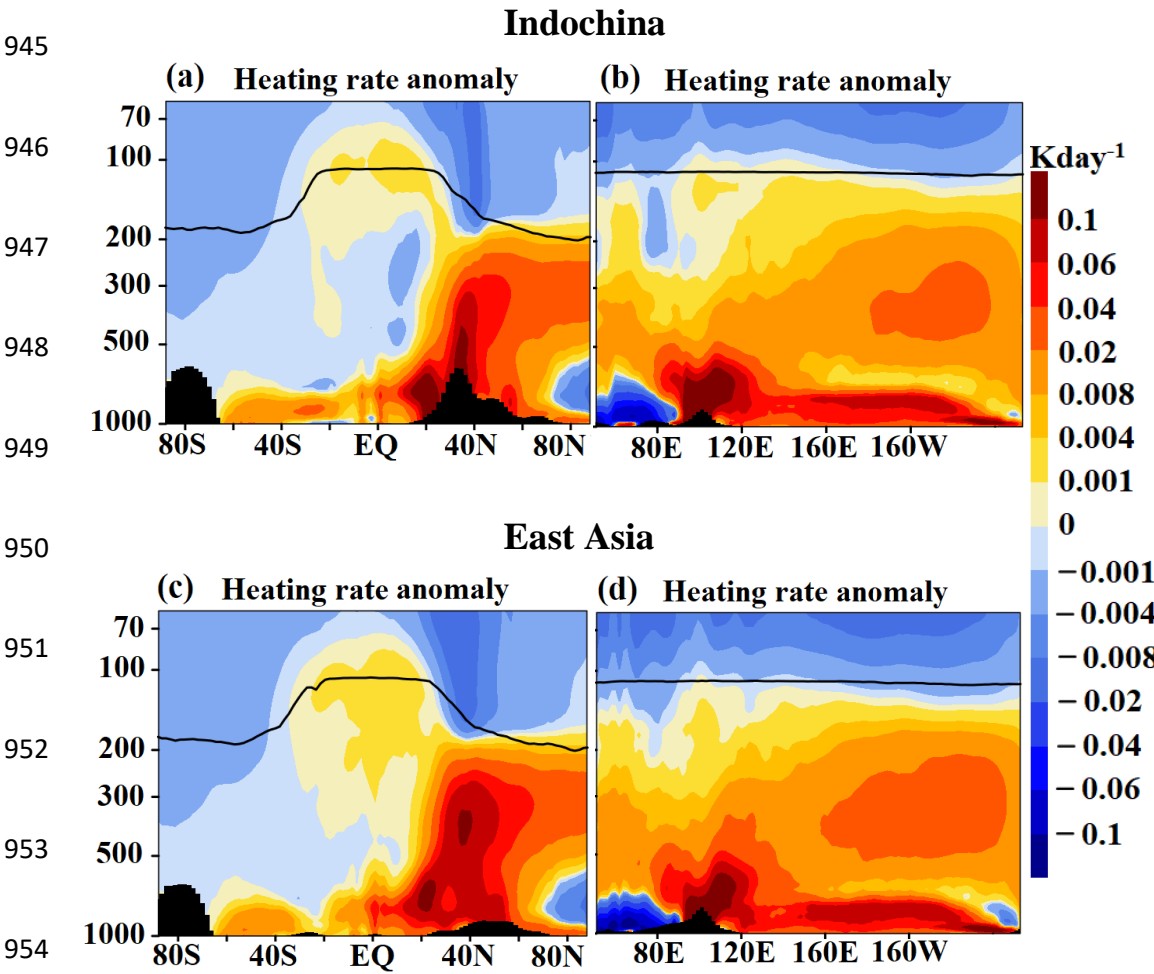

**Figure 8:** Vertical section of heating rate anomalies (K d$^{-1}$) for spring season 2013 from ECHAM6-HAMMOZ simulations (BMaeroon - BMaerooff) (a) latitude-pressure section averaged for 91°E - 107$^0$E, (b) longitude-pressure section averaged for 18$^0$N - 24$^0$N. (c) same as (a) but averaged for 108°E - 123$^0$E. (d) same as (b) but averaged for 22$^0$N - 27$^0$N. The black vertical bars show the topography and the black line indicates the tropopause.

965

966

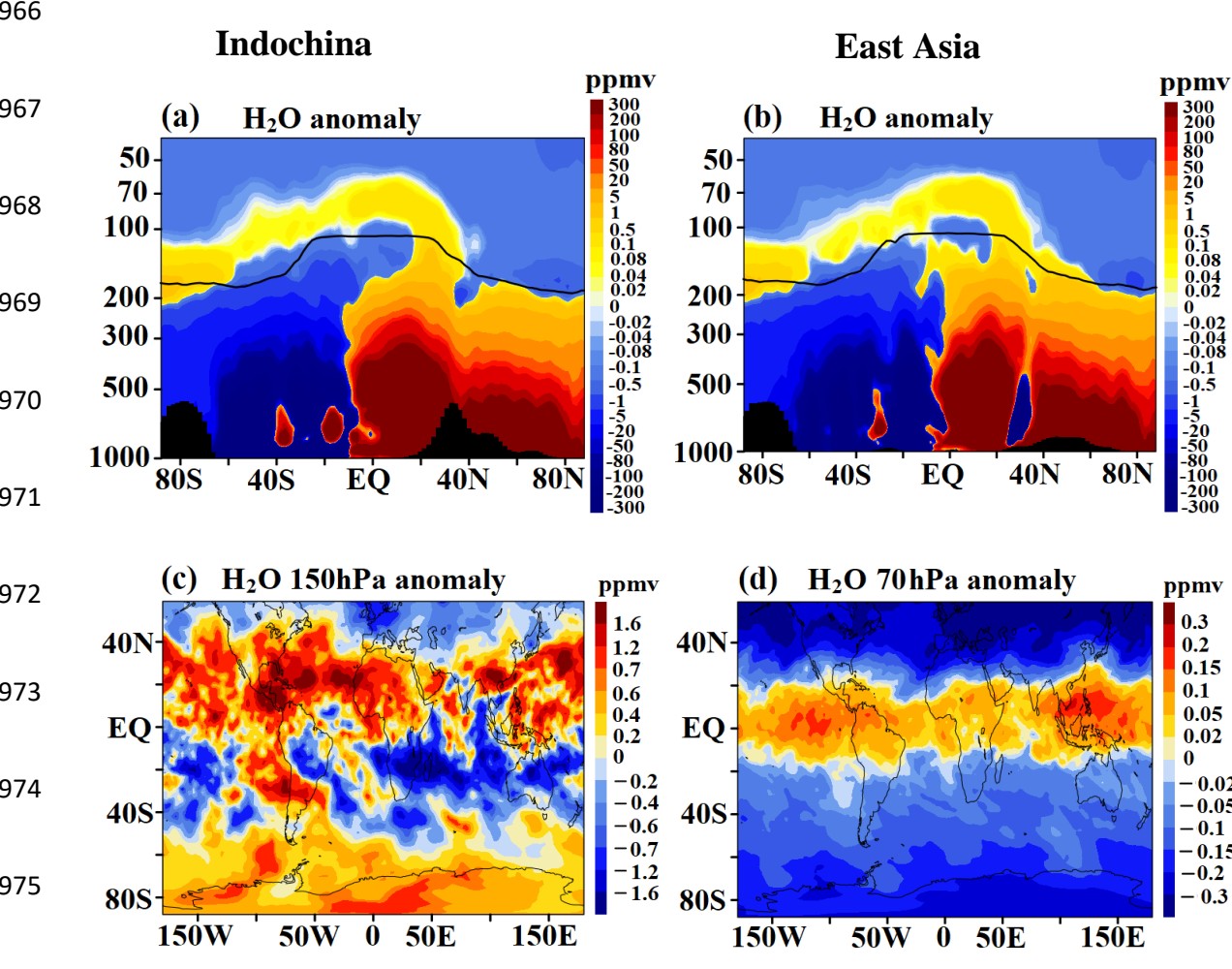

**Figure 9:** Vertical and horizontal distribution of anomalies of water vapour (ppmv) for spring
2013 from the ECHAM6-HAMMOZ simulations (BMaeroon - BMaerooff) (a) latitude-
pressure cross-section averaged for 91°E - 107°E, (b) longitude-pressure cross-section
averaged over 108°E-123°E, at (c) 150 hPa level, and (d) 70 hPa level. In Figure 9(a-b) the
black vertical bars show the topography and the black line indicates the tropopause.

