# Peer review of "The outflow of Asian biomass burning carbonaceous aerosol into the UTLS in spring"

_Atmospheric Chemistry and Physics, 2021_

## Referee Comment (RC2)

General comments:

The study by Chavan et al. discusses the transport pathway of Asian biomass burning aerosol to the UTLS during boreal spring and its radiative impacts using a CCM coupled an aerosol module. The topic is of great interest and of scientific importance. However, I have some doubts in the design of the experiments and the relative interpretations of the results, which will be mentioned in detail in the major comments. Besides, there are some missing information and improper presentations throughout the paper, which gave readers difficulty to understand. Thus, I would suggest the authors to reconsider the design of experiments and clarify the missing information before a more detailed revision. In general, I would suggest a major revision before publication.

Major comments:

Based on the description of configuration of two runs and "We performed 10-member ensemble runs … starting between 1 and 10 January 2012 and ending on 31 December 2013... ", I would guess that the BB aerosol should also be treated differently for two runs ("on" and "off") during the "spin-up" year 2012 (if I understand it wrong, please correct me). Therefore, the simulated aerosols and its effects during MAM 2013 should be combined and accumulated results of the whole simulation period beforehand. Here the aerosol transport time-scale from PBL to UT/LS and shallow BDC, the circulation changes related to the radiative effect from aerosol and water vapor as well as the complicated feedbacks should be all relevant. The whole paper is, to a large extend, misleading, which attributes the complex effects of BB aerosols only to the spring Asian BB.

Thus, one possible solution is to revise the design of experiments to isolate the effect of carbonaceous aerosols from Asian BB during the spring; another idea is to make some discussions that clarify the accumulated effects.

Minor comments:

1.  L105, some references or a website should be added for "AEROCOM-ACCMIP-II".

2.  L107-108, why there are three datasets for BB emissions (GICC, RETRO and GFED V2)? What is the indeed used in the simulations?

3. L113-114, the first level and second level is not an accurate description because the model level has not been introduced. "The first/second model level (~*hPa)" would be better.

4. L125-126, how did you vary SST and SIC? And for the sentence, I would suggest not to use "explore" since the effect from varying initial conditions are not discussed.

5. It seems to me that the Figure 3 (c) and (d) should be relevant: one is the AOD excluding the effect from dust; one is the AOD from carbonaceous aerosols. Could you comment on this point?

6. The description of longitude range in Figure 5 (85-140E) is not consistent with main body (L304).

7. L363, it should be "Fig. 6a-h".

Comments on Figures:

1. Figure 2 is not well organized. I would suggest : 1) add "BMaeroon" and "OSIRIS" to the titles of (e) and (f) respectively and use the same color bar; 2) plot the locations of the ten sites on the map in (c) and avoid the repeat of the longitudes and latitudes of them in the figure description since they are already mentioned in the paper; 3) the names of sites and their bars are not well aligned in subfigure (d).

2. Figure 3: 1) why the range of map in (b) is different from others? 2) what are the vectors shown in (b), wind anomalies? and clearly the upright vector symbol is in a wrong scale. 3) I suggest unifying the colorbars.

3. Please specify what is the vectors in Figure 5 (c-d) and Figure 6 (a-h).

4. In Figure 5(c), the upwelling (if it is the meridional circulation from BMaeroon) can be found from 10N-20N, not exactly from 10S-10N. Why did you choose 10S-10N as the range for Figure 5 (d)?

5. "seasonal mean" are used in a lot description of figures. I suggest changing them to "spring mean (MAM)".

---

## Author Comment (AC1)

[Figure]

**Figure 2**: (a) Aerosol optical depth (AOD) averaged for spring 2013 from MODIS, (b) same as (a) but from MISR, (c) same as (a) but from ECHAM6 - HAMMOZ BMaeroon simulation. (d) Comparison of simulated AOD (from BMaeroon) averaged for spring 2013 with AERONET observations at Gandhi college (GC; 25.81°N - 85.12°E ), Kathmandu Bode (BD; 27.68°N -85.39°E), Lumbini (LU; 27.49°N-83.28°E), Dhaka University (DU; 23.72°N - 90.39°E), Myanmar(MY; 16.86°N-96.15°E), Nghia Do (ND; 21.04°N - 105.80°E), Silpakorn University (SU; 13.81°N-100.04°E), Ubon Ratchathani (UR; 15.24°N - 104.87°E), Vientiane (VI; 17.99°N-102.57°E), Hong Kong Poly (HKP; 22.30°N – 114.18°E). (e) Simulated (BMaeroon) aerosol extinction coefficient (865 nm) (km$^{-1}$), averaged for 12°N -30°N and spring 2013 (f) same as (e) but from OSIRIS measurements (750 nm). White contours in Fig (a)-(c) indicate the orography (km) of the Tibetan Plateau.

[Figure]

**Figure 3**: Distribution of ECHAM6-HAMMOZ simulated anomalies of (BMaeroon - BMaerooff) (a) AOD, (b) dust AOD, (c) BC-AOD and OC-AOD, together, averaged for spring 2013. Streamlines in figure 3b indicate wind anomalies at 900 hPa (BMaeroon-BMaerooff).

[Figure]

**Figure 5**: (a) Distribution of Outgoing Longwave Radiation (OLR) (W m$^{-2}$) from NCEP reanalysis-2 data averaged for spring 2013, (b) same as (a) but from the ECHAM6-HAMMOZ simulations (BMaeroon). Vertical distribution of cloud droplet number concentration (CDNC) and ice crystal number concentration (ICNC) (1 mg$^{-1}$) averaged for spring 2013 from ECHAM6-HAMMOZ simulations (BMaeroon) (c) latitude-pressure section (average for $85^0$E - $140^0$E) and (d) longitude-pressure section (average for $10^oN$ - $20^oN$). Vectors of the circulation (BMaeroon) are shown in (c)-(d) with the vertical velocity field scaled by 300.

[Figure]

**Figure 6**: Vertical cross-section of anomalies of BC (ng m$^{-3}$) (BMaeroon – Bmaerooff) averaged for the spring 2013 and (a) latitude-pressure section (averaged for 91$^o$E-107$^o$E), (b) longitude-pressure section (averaged for 18$^o$N-24$^o$N). (c-d) is the same as (a-b) but for OC. (e) same as (a) but averaged over 108$^o$E-123$^o$E, (f) same as (b) but averaged for 18$^o$N-24$^o$N. (g-h) same as in (e-f) but for OC. The arrows in (a-h) indicate winds in m s$^{-1}$ with the vertical velocity field scaled by 300. The black vertical bar shows the topography and the black line indicates the tropopause.

[Figure]

**Figure S2**: Distribution of anomalies of dust aerosol (µg.m-3) (BMaeroon-BMaerooff) averaged for spring 2013 for (a) the lower troposphere (1000 to 700 hPa) and (b) the mid-upper troposphere (600 hPa - tropopause). Gray shading in Fig (a) indicates the Tibetan Plateau.

[Figure]

**Figure S3**: Distribution of anomalies (BMaeroon - BMaerooff) averaged for spring 2013 (a) atmospheric column concentration of BC and OC together (%), (b) ratio of BC-AOD to the total AOD (%), (c) ratio of OC-AOD to total AOD (%).

---

## Author Response (AR1)

**Replies to Reviewer-I**

**General comments:**

The paper, basing on model simulations validated with satellite and ground observations, investigates the transport of carbonaceous BB to the Upper Troposphere – Lower Stratosphere during spring and its impact on the radiative balance. The manuscript is presented in a clear and well-structured manner, and the topic is of relevance. I therefore encourage its publication, provided some minor revisions.

Reply: We thank the reviewer for valuable suggestions and appreciating our efforts. The suggestions made by reviewer are incorporated in the revised manuscript. The changes are indicated in blue color in the revised manuscript and corresponding line numbers are also mentioned in the replies given below.

**Specific comments:**

(1)     Lines 36-37: "It is one of the major sources of a large carbonaceous aerosol loading" This sentence needs to be more specific/referenced.

Reply(1): We have included a reference (Ni et al., 2019) now at L37.

(2)     Lines 69-70: Are those all the references for the projects? It seems like one is missing.

Reply(2): References for all the projects are included now (L72-73).

(3)Lines 105: is there any reference for AEROCOM-ACCMIP-II?

Reply(3): A reference for AEROCOM-ACCMIP-II is now included at L116.

(4) Lines 107-108: GICC, RETRO and GFED v2: are they summed/averaged? How are they used all together?

Reply(4): Sorry for not being clear. The ACCMIP-II emissions are derived from a harmonization of data from GICC (Mieville et al. 2010), RETRO (Schultz et al., 2008), and GFED v2 (Van Der Werf et al., 2006)." However, to avoid confusion it is removed now. The text is re-written as "The anthropogenic and fire emissions were obtained from the ACCMIP-II (Emissions for Atmospheric Chemistry and Climate Model Intercomparison Project) emission inventories and are interpolated for the period 2000-2100 by using Representative Concentration Pathway 4.5 (RCP4.5) (Lamarque et al., 2010; van Vuuren et al., 2011). (L112-116).

(5)Lines 109-110: Many inventories are representative only of averages over specific periods: is there a reason why forest and grass fires emissions are mentioned specifically in this respect?

Reply (5): We agree that many inventories are representative only of averages over specific periods. There is no need to mention forest and grass fires emissions explicitly and therefore this is removed now. The above sentence is rewritten as "The biomass burning emissions dataset represent average conditions of the decade (Tegen et al., 2019)." (L116-117)

(6) Lines 117: Is it necessary to mention sea salt and dust?

Reply(6): It is removed now.

(7) Lines 125-126: It is not clear how the setup of the simulations were made: what is meant in particular by "starting between 1 and 10 January 2012 and ending on 31 December 2013 to explore the variability due to the initial conditions."?

In which way do you vary the initial conditions? Are those variations the same between the members of the BMaerooff and the of BMaeroon?

Reply(7): Sorry for not being clear. It is re-written as "We performed two sets of emission sensitivity experiments; in one set of the simulations, the aerosol emissions from biomass burning were kept on (referred to as BMaeroon simulations) and in another set of the simulations, the aerosol emissions from biomass burning were kept off (referred to as BMaerooff simulations). We adopted an ensemble mean approach (with ten ensemble members) for the above two experiments. Ten spin-up simulations were performed from 1-10 January 2012 up to 28 February 2013 to generate stabilized initial fields for the ten ensemble members. Emissions were the same in each of the ten members during the spin-up period. In the BMaerooff simulations (ten ensemble members each), the biomass burning aerosols were switched off since 1 March 2013. The BMaeroon and BMaerooff simulations ended on 31 December 2013. To investigate the effects of biomass burning aerosols emissions in spring (i.e., since 1 March 2013), we analyze the difference between BMaeroon and BMaerooff simulations for the spring season in 2013. (L132-144).

(8)Lines 141-143: The fire counts product is not properly presented and it remains unclear what does the mcd14d variable is representing. Where does it come from? Can the author put a summary sentence about the basics on how these values are estimated?

Reply(8): We have now explained that the variable mcd14d represents MODIS combined Terra/Aqua daily fire location data. We have now explained how fire values are estimated as below (L156-165).

In order to study spatio-temporal variations in the biomass burning activity, we analysed the Terra/Aqua combined daily active fire location data (product mcd14dl) from the Moderate

Resolution Imaging Spectroradiometer (MODIS) (https://firms.modaps.eosdis.nasa.gov/download/) onboard Terra and Aqua (Earth Observing System). This MODIS collection-6, Level-2 global data are processed by NASA's Land, Atmosphere Near real-time Capability for EOS (LANCE) Fire Information for Resource Management System (FIRMS), using swath products (MOD14/MYD14). The thermal anomaly / active fire represents the centre of a 1 km pixel that is flagged by the MODIS MOD14/MYD14 Fire and Thermal Anomalies algorithm as containing one or more fires within the pixel (Giglio et al., 2003).

(9) Lines 210-211: Is there any existing inter-comparison study that explains/show such difference between the two products?

Reply(9): Thank you for the suggestion. We have mentioned the inter-comparison study that explains differences between MODIS and MISR as below.

The differences between MISR and MODIS may be due to differences in their calibration, algorithm assumptions, or the aerosol models in the lookup tables used in the retrieval algorithms (Addou et al., 2005; Choi et al., 2019). (L268-271)

(10) Fig 2: Is the amount of aerosol over the Himalaya indicated in the simulation expected to be a model overestimation, or can it be plausible? Why are there such low modelled values with respect to measurements in the region southerner than 10N? It seems quite a coherent pattern, is it related to any specific circulation feature?

Reply(10): The model shows overestimation of AOD over the Tibetan plateau region and underestimation over the Himalayas. Yes, there is a pattern showing underestimation to the south of 13°N and overestimation of AOD over central India. This is due to the high amount

of dust emission over west Asia that is transported to India (Fig. S2). We have added related discussions in the revised manuscript at L240-255.

The simulated AOD is underestimated south of 13°N compared to MISR and MODIS (MODIS: 0.4 to 0.7, MISR:0.4 to 0.6, model: 0.21 to 0.3) and overestimated over central India (lat: 20°-28°N lon: 75°E-88°E) compared to MODIS and MISR (MODIS:0.16 to 0.4, MISR:0.21 to 0.3, model: 0.3 to 0.5). These issues may be due to a higher amount of dust emission in the model over West Asia that is transported to India. In the past, a number of papers reported that transport of dust occurs from west Asia to the Indo-Gangetic plain and the Tibetan Plateau region during spring (Lau and Kim 2006; Fadnavis et al., 2017b, Fadnavis et al., 2021a). Simulated AOD is also overestimated over the Tibetan Plateau and East Asian region (MODIS: 0.21 to 1.0, MISR: 0.16 to 0.6, model: 0.27 to 1.2). The distribution of dust AOD also shows high amounts over these regions (See Fig. S1). This indicates that higher amounts of dust over the Tibetan Plateau and the East Asia region cause overestimation of AOD there. Tegen et al. (2019) also reported that in ECHAM6–HAMMOZ simulations the AOD is overestimated over East Asia in comparison with MISR. The model simulations underestimate the AOD over the Himalayas in comparison with MODIS (MODIS: 0.24 to 0.3, MISR: 0.1 to 0.21, model: 0.1 to 0.3).

(11)Lines 250-252: Please, justify this statement more in depth.

Reply(11): We have now elaborated it as "It shows enhanced AOD anomalies over the Indo-Gangetic plain (~0.22 to 0.8), the Tibetan Plateau and the north eastern parts of East Asia (~0.3 to 1.2). The distribution of anomalies in dust AOD shows high amounts over these regions. It indicates that dust enhancement over the Indo-Gangetic plain (~0.22 to 0.8), the Tibetan Plateau and the northeastern parts of East Asia (0.8 to 1) (Fig. 3b) causes enhancement in AOD there. The simulated dust anomalies and circulation patterns also show

transport of enhanced dust from West Asia to North India and the Indo-Gangetic plain region in the lower troposphere (Fig. 3b and Fig. S2a). Dust is also transported from Tibetan Plateau-East Asia region to North India in the mid/upper troposphere (Fig. S2b). The enhanced dust transport from west Asia and Tibetan Plateau-East Asia region to South Asia is induced by atmospheric heating generated by biomass burning carbonaceous aerosols (discussed in section 4.4). This atmospheric heating leads to enhance dust emission over the respective desert regions. Dust being absorptive in nature contributes to a further increase of the atmospheric heating. The heating led to a formation of a low pressure zone over East India in the lower troposphere (900 hPa) (Fig. 3b) and the Bay of Bengal and Myanmar in the mid-troposphere (500 hPa) (Fig. S2b and Fig. 7b). These circulation changes further enhanced the dust transport from west Asia and the Tibetan Plateau-East Asia region to South Asia." (L290-306).

(12)Lines 381-382: This sentence is too generically formulated

Reply(12): We show anomalies in vertical velocity to justify the above sentence. The above sentence is now re-written as: The cross tropopause transport is reinforced by enhanced vertical motion (Fig. S6a-b) produced by the heating generated by the carbonaceous aerosols. L440-442.

(12)Line 411: This is the first time in the whole manuscript that the ENSO is mentioned. A) It is worth to have a short indication of the reason why this is relevant for the presented study B) If kept, since it is the only time it is referred, the acronym has to be rewritten in full extension.

Reply(12): We have removed the abbreviation at L411. It is now referred at L147. As suggested we have given its relevance to the present study at L146-149. It is re-written as "The year 2013 was chosen for the analysis as this was a neutral year without a pronounced

El Niño or Indian Ocean Dipole oscillation. Such large-scale coupled atmosphere–ocean oscillations substantially affect the transport processes to the UTLS (Fadnavis et al., 2017a, 2019).

(13) Lines 421-426: those two sentences are somehow a repetition.

Reply(13): We have avoided the repetition. It is re-written as "Our analysis shows that deep convection, which occurs over the Malay peninsula and Indonesia, transports carbonaceous aerosols from the boundary layer of the Indochina and East Asia region into the lowermost stratosphere (BC: 0.1 to 6 ng m$^{-3}$ for BC, OC: 0.2 to 10 ng m$^{-3}$)." at L480-483.

(14) Lines 454-455: It would be useful here for the reader to have a more quantitative way to understand what "counterbalanced" means. Which are the typical values of stratospheric cooling by CO2?

Reply(14): It is known that 355 ppm CO2 produces a cooling of 1 - 13.0 K day$^{-1}$ in the stratosphere that peaks at the stratopause (Clough and Iacono, 1995)." Impact of stratospheric water vapour on stratospheric temperature is complex since it may vary with latitude/altitude, e.g. Wang et al., (2020) report that increase of water vapor in tropical and subtropical lower stratosphere cause warming of the lower stratosphere, offsetting the cooling caused by doubling of $CO_2$. Maycock et al. (2013) found that a uniform doubling in stratospheric water vapor causes stratospheric cooling with a maximum amplitude of 5–6 K in the polar lower stratosphere and 2–3 K in the tropical lower stratosphere. Solomon et al. 2010 and Foster & Shine (1999) report that stratospheric water vapour radiatively cools the stratosphere. Discussion on this topic is out of the scope of the present paper. However, the impact of stratospheric water vapour on climate is certain: it causes warming of the climate (Banerjee et al., 2019; Dessler et al., 2013). Hence when reporting the climatic impact of

stratospheric water vapor we have re-written above sentence as "An increase in stratospheric water vapour is important as it has an impact on stratospheric temperatures and thus indirectly on stratospheric dynamics (Maycock et al., 2013). The moistening of the stratosphere produces a positive feedback on the climate (Banerjee et al., 2019; Dessler et al., 2013). (L510-513).

References:

Wang, T. Q.,  Zhang, M., Kuilman, and A., Hannachi.: "Response of Stratospheric Water Vapour to CO2 Doubling in WACCM." Clim. Dyn. 54(11–12):4877–89, 2020.

Banerjee, A.,  Chiodo, G., Previdi, M.,  Ponater, M., Conley, A. J., Polvani, L. M.: Stratospheric water vapor: an important climate feedback, Clim. Dyn, 53:1697–1710 https://doi.org/10.1007/s00382-019-04721-4, 2019.

Desslera A. E., Schoeberl, M. R., Wanga , T., Davis, S. M., and Rosenlof, K. H.: Stratospheric water vapor feedback, PNAS 110,  18087–18091, 2013.

Maycock, A. C.,  JoshiM. M.,  Shine, K. P., Scaife,  A. A.: The Circulation Response to Idealized Changes in Stratospheric Water Vapor, J. Clim., 26,  545-561, DOI: 10.1175/JCLI-D-12-00155.1, 2013.

**Technical comments:**

(15) Fig.2 is a bit messy and non-coherent in all its panels' layout (sizes and position of panels and fonts). This is for example particularly evident when trying to compare the results of panels e and f. Also, it will be useful to specify in the title of those two panels which one is BMaeroon and which is from OSIRIS. Moreover, it will be better to have the same color scale. I would also suggest making the caption lighter, moving the information with the location of the different stations in a separated table.

Reply(15): As suggested, Figure 2 is replotted. The locations of the10 sites are shown in Fig 2c. We have used abbreviations for the locations of the ten sites in Fig 2d. The details of abbreviations and locations of the ten sites (latitude and longitude) are described in the figure caption.

[Figure]

Figure 2: (a) Aerosol optical depth (AOD) averaged for spring 2013 from MODIS, (b) same as (a) but from MISR, (c) same as (a) but from ECHAM6 - HAMMOZ BMaeroon simulation. White contours in Fig (a)-(c) indicate the orography (km) of the Tibetan Plateau. (d) Comparison of simulated AOD (from BMaeroon) averaged for spring 2013 with AERONET observations at Gandhi college (GC; 25.81°N - 85.12°E ), Kathmandu Bode (BD; 27.68°N -85.39°E), Lumbini (LU; 27.49°N-83.28°E), Dhaka University (DU; 23.72°N - 90.39°E), Myanmar(MY; 16.86°N-96.15°E), Nghia Do (ND; 21.04°N - 105.80°E), Silpakorn University (SU; 13.81°N-100.04°E), Ubon Ratchathani (UR; 15.24°N - 104.87°E), Vientiane (VI; 17.99°N-102.57°E), Hong Kong Poly (HKP; 22.30°N – 114.18°E). (e) Simulated (BMaeroon) aerosol extinction coefficient (865 nm) (km$^{-1}$), averaged for 12ºN-30ºN and spring 2013 (f) same as (e) but from OSIRIS measurements (750 nm).

(16) Fig.3: Is there any specific reason why the panel b, with the DUST anomaly, is not in the same domain as the others? It would be very useful to have the same, also, since we are missing otherwise the information on the high dust aod contribution over East-China that seems to be suggested from panel c. What is the purpose of the wind arrows of figure 3b?

Reply(16): We show transport of dust from west Asia to India and its influence on AOD. Wind vectors are plotted to show the circulation pattern. Hence the domain for Fig. 3b was different than for the rest for the panels in Fig.3. As suggested, we have extended this figure over the East-China region. The discussion on transport of dust is made clear at L290-306 as below:

"It shows enhanced AOD anomalies over the Indo-Gangetic plain (~0.22 to 0.8), the Tibetan Plateau and the north eastern parts of East Asia (~0.3 to 1.2). The distribution of anomalies in dust AOD shows high amounts over these regions. It indicates that dust enhancement over the Indo-Gangetic plain (~0.22 to 0.8), the Tibetan Plateau and the northeastern parts of East Asia (0.8 to 1) (Fig. 3b) causes enhancement in AOD there. The simulated dust anomalies and circulation patterns also show transport of enhanced dust from West Asia to North India and the Indo-Gangetic plain region in the lower troposphere (Fig. 3b and Fig. S2a). Dust is also transported from Tibetan Plateau-East Asia region to North India in the mid/upper troposphere (Fig. S2b). The enhanced dust transport from west Asia and Tibetan Plateau-East Asia region to South Asia is induced by atmospheric heating generated by biomass burning carbonaceous aerosols (discussed in section 4.4). This atmospheric heating leads to enhance dust emission over the respective desert regions. Dust being absorptive in nature contributes to a further increase of the atmospheric heating. The heating led to a formation of a low pressure zone over East India in the lower troposphere (900 hPa) (Fig. 3b) and the Bay of Bengal and Myanmar in the mid-troposphere (500 hPa) (Fig. S2b and Fig. 7b). These

circulation changes further enhanced the dust transport from west Asia and the Tibetan Plateau-East Asia region to South Asia.

(17) S1: The caption should specify that those are relative anomalies

Reply(17): We have defined anomalies as BMaeroon – Bmaerooff. It is mentioned in the caption of Fig. S1 which is now Fig. S3.

[Figure]

Figure S3: Distribution of anomalies (BMaeroon - BMaerooff) averaged for spring 2013 (a) atmospheric column concentration of BC and OC together (%), (b) ratio of BC-AOD to the total AOD (%), (c) ratio of OC-AOD to total AOD (%).

(18) Line 285: ..reported A radiative forcing…

Reply(18): It is corrected now at L336

(19) Figure 5: I would strongly suggest to report panel b and panel d in the same longitude range and size, to easily individuate the regions of uplift.

Reply(19): It is correct now (added below)

[Figure]

**Figure 5**. (a) Distribution of Outgoing Longwave Radiation (OLR) (W m$^{-2}$) from NCEP reanalysis-2 data averaged for spring 2013, (b) same as (a) but from the ECHAM6-HAMMOZ simulations (BMaeroon). Vertical distribution of cloud droplet number concentration (CDNC) and ice crystal number concentration (ICNC) (1 mg$^{-1}$) averaged for spring 2013 from ECHAM6-HAMMOZ simulations (BMaeroon) (c) latitude-pressure section (average for 85$^0$E-140$^0$E) and (d) longitude-pressure section (average for 10$^o$N - 20$^o$N). Vectors of the circulation (BMaeroon) are shown in (c)-(d) with the vertical velocity field scaled by 300.

(20) Figure 6: Please adjust the size of the figures, which looks vertically stretched.

Reply(20): Figure 6 is replotted.

[Figure]

**Figure 6**: Vertical cross-section of anomalies of BC (ng m⁻³) (BMaeroon – Bmaerooff) averaged for spring 2013 (a) latitude-pressure section (averaged for 91ºE-107ºE), (b) longitude-pressure section (averaged for 18ºN-24ºN). (c-d) is the same as (a-b) but for OC. (e) same as (a) but averaged over 108ºE-123ºE, (f) same as (b) but averaged for 18ºN-24ºN. (g-h) same as in (e-f) but for OC. The arrows in (a-h) indicate winds in m s⁻¹ with the vertical velocity field scaled by 300. The black vertical bars show the topography and the black line indicates the tropopause.

**Replies to Reviewer-II**

General comments:

The study by Chavan et al. discusses the transport pathway of Asian biomass burning aerosol to the UTLS during boreal spring and its radiative impacts using a CCM coupled an aerosol module. The topic is of great interest and of scientific importance. However, I have some doubts in the design of the experiments and the relative interpretations of the results, which will be mentioned in detail in the major comments. Besides, there are some missing information and improper presentations throughout the paper, which gave readers difficulty to understand. Thus, I would suggest the authors to reconsider the design of experiments and clarify the missing information before a more detailed revision. In general, I would suggest a major revision before publication.

Reply: We thank the reviewer for valuable suggestions. We have now clarified the details of the design of the experiment in the revised manuscript (L132-144). We have incorporated other suggestions given by the reviewer. The changes are indicated in blue color in the revised manuscript and the corresponding line numbers are also mentioned in the replies given below.

Major comments:

(1)     Based on the description of configuration of two runs and "We performed 10-member ensemble runs  starting between 1 and 10 January 2012 and ending on 31 December 2013... ", I would guess that the BB aerosol should also be treated differently for two runs ("on" and "off") during the "spin-up" year 2012 (if I understand it wrong, please correct me). Therefore,

the simulated aerosols and its effects during MAM 2013 should be combined and accumulated results of the whole simulation period beforehand. Here the aerosol transport time-scale from PBL to UT/LS and shallow BDC, the circulation changes related to the radiative effect from aerosol and water vapor as well as the complicated feedbacks should be all relevant. The whole paper is, to a large extend, misleading, which attributes the complex effects of BB aerosols only to the spring Asian BB. Thus, one possible solution is to revise the design of experiments to isolate the effect of carbonaceous aerosols from Asian BB during the spring; another idea is to make some discussions that clarify the accumulated effects.

Reply(1): Sorry for not being clear in describing our model runs, which led to doubts on the experimental design. In the revised version, we clarify that in our experiments, emissions were the same during the spin-up period. We have now re-written it at L132-144 as below:

"We performed two sets of emission sensitivity experiments; in one set of the simulations, the aerosol emissions from biomass burning were kept on (referred to as BMaeroon simulations) and in another set of the simulations, the aerosol emissions from biomass burning were kept off (referred to as BMaerooff simulations). We adopted an ensemble mean approach (with ten ensemble members) for the above two experiments. Ten spin-up simulations were performed from 1-10 January 2012 up to 28 February 2013 to generate stabilized initial fields for the ten ensemble members. Emissions were the same in each of the ten members during the spin-up period. In the BMaerooff simulations (ten ensemble members each), the biomass burning aerosols were switched off since 1 March 2013. The BMaeroon and BMaerooff simulations ended on 31 December 2013. To investigate the effects of biomass burning aerosol emissions in spring (i.e., since 1 March 2013), we analyze the difference between BMaeroon and BMaerooff simulations for the spring season in 2013. "

In the past, many different studies have used different periods for spin-up (six-month to one year). For example, the study by Kokkola et al., 2018 used a 1-year spin-up period. Bergman et al. (2008) used a spin-up spanning six months. Also, Vozella et al (2012) used a one year spin-up period.

Kokkola et al., 2018, SALSA2.0: The sectional aerosol module of the aerosol–chemistry–climate model ECHAM6.3.0-HAM2.3-MOZ1.0, Geosci. Model Dev., 11, 3833–3863, 2018.

Vozella et al : Aerosol optical depth over the Arctic:A comparison of ECHAM-HAM and TM5 with ground-based, satellite and reanalysis data, Atmospheric Chemistry and Physics 12(15):8319-8353, DOI: 10.5194/acp-12-6953-2012, 2012>

Bergman et al., 2008, Evaluation of the sectional aerosol microphysics module SALSA implementation in ECHAM5-HAM aerosol-climate model, Geosci. Model Dev., 5, 845–868, 2012

Minor comments:

(1) L105, some references or a website should be added for "AEROCOM-ACCMIPII".

 Reply(1) : As suggested we have added a reference for AEROCOM-ACCMIPII at L116.

(2) L107-108, why there are three datasets for BB emissions (GICC, RETRO and GFED V2)? What is the indeed used in the simulations?

Reply(2): We have used emission from ACCMIP-II. The ACCMIP-II emissions are derived from a harmonization of data from GICC (Mieville et al. 2010), RETRO (Schultz et al., 2008), and GFED v2 (Van Der Werf et al., 2006). However, to avoid confusion it is removed now. The text is re-written as "The anthropogenic and fire emissions were obtained from the ACCMIP-II (Emissions for Atmospheric Chemistry and Climate Model Intercomparison Project) emission inventories and are interpolated for the period 2000-2100 by using Representative Concentration Pathway 4.5 (RCP4.5) (Lamarque et al., 2010; van Vuuren et al., 2011). ( L112-116).

(3) L113-114, the first level and second level is not an accurate description because the model level has not been introduced. "The first/second model level (~*hPa)" would be better.

Reply(3) : Here, "First level" indicates the model's first level above the boundary layer. The second level indicates the model's second level above the boundary layer. However, the height of the boundary layer varies therefore the first level or the second level are not at fixed pressure levels. However, the above sentence is now re-written as below at L118-122. Injection heights of biomass burning emissions are documented by Val Martin et al. (2010). The majority (75%) of the emissions are evenly distributed within the planetary boundary layer (PBL) with 17% in the first model level above the planetary boundary layer and 8% in the second model level above the planetary boundary layer (Tegen et al., 2019).

(4) L125-126, how did you vary SST and SIC? And for the sentence, I would suggest not to use "explore" since the effect from varying initial conditions are not discussed.

Reply(4) We have used monthly varying SST and SIC. It is now mentioned as "Atmospheric Model Inter-comparison Project (AMIP) monthly varying sea surface temperature (SST) and sea ice cover (SIC) were used as lower boundary conditions. (L130-L132).

(5) It seems to me that the Figure 3 (c) and (d) should be relevant: one is the AOD excluding the effect from dust; one is the AOD from carbonaceous aerosols. Could you comment on this point?

Reply(5): Figure 3 (c) includes water soluble aerosols in addition to carbonaceous aerosols. However, it does not provide any additional information, hence it is removed now. We have shown the distribution of dust in the lower (Fig. S3a) and mid-upper troposphere (Fig. S3b) together with the associated circulation to show the transport of dust.

(6) The description of longitude range in Figure 5 (85-140E) is not consistent with main body (L304).

Reply: It is corrected now as 85°-140°E at L356.

(7) L363, it should be "Fig. 6a-h".

Reply(7): It is corrected now.

Comments on Figures:(8) Figure 2 is not well organized. I would suggest : 1) add "BMaeroon" and "OSIRIS" to the titles of (e) and (f) respectively and use the same color bar; 2) plot the locations of the ten sites on the map in (c) and avoid the repeat of the longitudes and latitudes of them in the figure description since they are already mentioned in the paper; 3) the names of sites and their bars are not well aligned in subfigure (d).

Reply(8): Figure 2 is replotted. As suggested, we have shown the location of the ten sites in Fig. 2c. We have used abbreviations for locations of the ten sites in Fig. 2d. The abbreviations are described in the figure caption (shown below)

[Figure]

Figure 2: (a) Aerosol optical depth (AOD) averaged for spring 2013 from MODIS, (b) same as (a) but from MISR, (c) same as (a) but from ECHAM6 - HAMMOZ BMaeroon simulation. (d) Comparison of simulated AOD (from BMaeroon) averaged for spring 2013 with AERONET observations at Gandhi college (GC; 25.81°N - 85.12°E ), Kathmandu Bode (BD; 27.68°N -85.39°E), Lumbini (LU; 27.49°N-83.28°E), Dhaka University (DU; 23.72°N - 90.39°E), Myanmar(MY; 16.86°N-96.15°E), Nghia Do (ND; 21.04°N - 105.80°E), Silpakorn University (SU; 13.81°N-100.04°E), Ubon Ratchathani (UR; 15.24°N - 104.87°E), Vientiane (VI; 17.99°N-102.57°E), Hong Kong Poly (HKP; 22.30°N – 114.18°E). (e) Simulated (BMaeroon) aerosol extinction coefficient (865 nm) (km$^{-1}$), averaged for 12°N -30°N and spring 2013 (f) same as (e) but from OSIRIS measurements (750 nm). White contours in Fig (a)-(c) indicate the orography (km) of the Tibetan Plateau.

(9) Figure 3: 1) why the range of map in (b) is different from others? 2) what are the vectors shown in (b), wind anomalies? and clearly the upright vector symbol is in a wrong scale. 3) I suggest unifying the colorbars.

Reply(9) : We show transport of dust from west Asia to India and its influence on AOD. Wind anomalies are shown as streamlines to show the circulation pattern. It is discussed in the manuscript at L281-297. Hence the domain for Fig. 3b was different than rest of the panels. However, now it is made the same as rest of the panels in Fig.3.

As suggested, the same colorbar is now used for Fig 3(a) AOD and 3(b) dust AOD. The values of total AOD, dust AOD are higher than BC+OC AOD. Hence, we used different scales of colour bar for BC+OC AOD to show regions of high carbonaceous aerosols. This feature is suppressed if we use a unified colour bar.

[Figure]

**Figure 3**. Distribution of ECHAM6-HAMMOZ simulated anomalies of (BMaeroon - BMaerooff) (a) AOD, (b) dust AOD, (c) BC-AOD and OC-AOD, together, averaged for spring 2013. Streamlines in figure 3b indicate wind anomalies at 900 hPa (BMaeroon-BMaerooff).

(10) Please specify what is the vectors in Figure 5 (c-d) and Figure 6 (a-h).

Reply(10): The vectors in Figure 5 (c-d) and Figure 6 (a-h) indicate circulation. It is now mentioned in the caption of Figure 5 (c-d) and Figure 6 (a-h). (shown below)

[Figure]

**Figure 5**. (a) Distribution of Outgoing Longwave Radiation (OLR) (W m$^{-2}$) from NCEP reanalysis-2 data averaged for spring 2013, (b) same as (a) but from the ECHAM6-HAMMOZ simulations (BMaeroon). Vertical distribution of cloud droplet number concentration (CDNC) and ice crystal number concentration (ICNC) (1 mg$^{-1}$ ) averaged for spring 2013 from ECHAM6-HAMMOZ simulations (BMaeroon) (c) latitude-pressure section (average for $85^0$E-$140^0$E) and (d) longitude-pressure section (average for $10^oN$ - $20^oN$). Vectors of the circulation (BMaeroon) are shown in (c)-(d) with the vertical velocity field scaled by 300.

[Figure]

**Figure 6.** Vertical cross-section of anomalies of BC (ng m$^{-3}$) (BMaeroon – Bmaerooff) averaged for the spring 2013 and (a) latitude-pressure section (averaged for 91ºE-107ºE), (b) longitude-pressure section (averaged for 18ºN-24ºN). (c-d) is the same as (a-b) but for OC. (e) same as (a) but averaged over 108ºE-123ºE, (f) same as (b) but averaged for 18ºN-24ºN. (g-h) same as in (e-f) but for OC. The arrows in (a-h) indicate winds in m s$^{-1}$ with the vertical velocity field scaled by 300. The black vertical bar shows the topography and the black line indicates the tropopause.

(11) In Figure 5(c), the upwelling (if it is the meridional circulation from BMaeroon) can be found from 10N-20N, not exactly from 10S-10N. Why did you choose 10S-10N as the range for Figure 5 (d)?

Reply(11): Thank you for the suggestion. We have now shown upwelling averaged over 10°-20°N. (Shown above in reply 2)

(12) "seasonal mean" are used in a lot description of figures. I suggest changing them to "spring mean (MAM)".

Reply(12): We have modified it as averaged for spring 2013 (L330, L817, L838, L841, L853-854, L880-881, L902, L904-905).

---

## Author Response (AR2)

Replies to Technical corrections:

We thank the Editor for appreciating our efforts. The changes are indicated in blue color in the revised manuscript and corresponding line numbers are also mentioned in the replies given below.

P6, L133: It would be nice if you could motivate somewhere in this section why you use an ensemble simulation.

Reply: Thank you for the suggestion. We have now added a reason for using an ensemble mean approach at L136-137, also given below.

"To assess the uncertainty caused by model imperfections, we adopted an ensemble mean approach (with ten ensemble members) for the above two experiments."

P6, L146: Instead of just writing "different initial conditions" I would appreciate if you could give more details on how the ensemble members were initialised and what the difference in initialisation of these are.

Reply: As suggested, we have given the details of how the ensemble members were initialised and initial conditions used at L145-149, also given below.

The uncertainty estimates in simulated radiative forcing, heating rates, and aerosol extinction coefficient are obtained from the difference between the mean of (a) the ten-members for BMaeroon and (b) the ten-members for BMaerooff. Both sets were generated from initial conditions with start times shifting by a day over the ten days period of 1-10 January.

P17, L421: Add a comma after "Further"

Reply: It is corrected.

P18, L447: shows -> show (?)

Reply: It is corrected

P35, L845: space between "Myanmar" and "(MY...)" is missing

Reply: It is corrected (now at L836).

P36, L859: figure 3 -> Figure 3

Reply: It is corrected (Now at L850).

P40, Figure 7 figure header: Space between K and day-1 is missing. Further, I think it should rather read K d-1, please check the Copernicus guidelines and correct if necessary throughout the manuscript.

Reply: It is corrected as K d$^{-1}$ at P40 and in the manuscript.

P40, Figure 7 caption, L958: remove dot between the units

Reply: It is corrected now.

Supplement, Figure S2 caption: Same here, remove dot between the units.

Reply: It is corrected now.